# IL-33-mediated mast cell activation promotes gastric cancer through macrophage mobilization

Moritz F. Eissmann[1], Christine Dijkstra[1], Andrew Jarnicki[10], Toby Phesse [1,6], Jamina Brunnberg[1,7], Ashleigh R. Poh[1], Nima Etemadi [1,8], Evelyn Tsantikos[2], Stefan Thiem[1], Nicholas D. Huntington[3], Margaret L. Hibbs[2], Alex Boussioutas[4], Michele A. Grimbaldeston[5,9], Michael Buchert[1], Robert J.J. O'Donoghue[1,10], Frederick Masson[1,11] & Matthias Ernst [1]

The contribution of mast cells in the microenvironment of solid malignancies remains controversial. Here we functionally assess the impact of tumor-adjacent, submucosal mast cell accumulation in murine and human intestinal-type gastric cancer. We find that genetic ablation or therapeutic inactivation of mast cells suppresses accumulation of tumor-associated macrophages, reduces tumor cell proliferation and angiogenesis, and diminishes tumor burden. Mast cells are activated by interleukin (IL)-33, an alarmin produced by the tumor epithelium in response to the inflammatory cytokine IL-11, which is required for the growth of gastric cancers in mice. Accordingly, ablation of the cognate IL-33 receptor St2 limits tumor growth, and reduces mast cell-dependent production and release of the macrophage-attracting factors Csf2, Ccl3, and Il6. Conversely, genetic or therapeutic macrophage depletion reduces tumor burden without affecting mast cell abundance. Therefore, tumor-derived IL-33 sustains a mast cell and macrophage-dependent signaling cascade that is amenable for the treatment of gastric cancer.

[1] Cancer and Inflammation Laboratory, Olivia Newton-John Cancer Research Institute and School of Cancer Medicine, La Trobe University, Heidelberg, VIC 3084, Australia. [2] Department of Immunology and Pathology, Monash University, Melbourne, VIC 3004, Australia. [3] Molecular Immunology Division, The Walter and Eliza Hall Institute of Medical Research, and Department of Medical Biology, University of Melbourne, Melbourne, VIC 3052, Australia. [4] Department of Medicine, University of Melbourne, Melbourne, VIC 3050, Australia. [5] Centre for Cancer Biology, University of South Australia and SA Pathology, Adelaide, SA 5000, Australia. [6] Present address: Cell Signaling and Cancer Laboratory, European Cancer Stem Cell Research Institute and Cardiff University, Cardiff CF24 4HQ, UK. [7] Present address: Institute of Biochemistry, Goethe University Frankfurt, Frankfurt am Main, 60438 Frankfurt, Germany. [8] Present address: Cell Signalling and Cell Death Division, Walter and Eliza Hall Institute of Medical Research, and Department of Medical Biology, University of Melbourne, Melbourne, VIC 3052, Australia. [9] Present address: OMNI-Biomarker Development, Genentech Inc., South San Francisco, CA 94080, USA. [10] Present address: Department of Pharmacology and Therapeutics, University of Melbourne, Melbourne, VIC 3010, Australia. [11] Present address: Team 5, Centre of Physiopathology Toulouse-Purpan, INSERM UMR 1043/CNRS UMR 5282, University Toulouse III, CHU Purpan, 31024 Toulouse, France. Correspondence and requests for materials should be addressed to M.E. (email: Matthias.ernst@onjcri.org.au)

The interactions between cancer cells and their microenvironment can result in tumor progression as well as suppression and/or eradication of cancers[1]. Besides extracellular matrix and stromal cells, the tumor microenvironment is composed of immune cells of the adaptive and innate immune system, with the latter comprising neutrophils, macrophages, mast cells, myeloid-derived suppressor cells, dendritic cells, innate lymphocytes, and natural killer cells. Often T-lymphocytes and tumor-associated macrophages (TAMs) account for the most abundant immune cell populations infiltrating established tumors[2,3].

Mast cells are long-lived secretory cells of hematopoietic origin that function as sentinels by responding to changes in their environment[4]. They respond to extrinsic signals through a multitude of cell surface receptors to secrete histamine and proteases from prestored sources in cytoplasmic granules[5], alongside newly synthesized inflammatory mediators[4]. Accordingly, mast cells can be part of innate and adaptive immune responses and therefore contribute to various pathophysiological conditions[6]. In cancer, the presence of mast cells, both at the tumor periphery and in the tumor core can correlate with disease progression, increased metastasis and reduced survival of patients with melanoma[7], prostate cancer[8], pancreatic adenocarcinoma[9], squamous cell carcinoma[10], and gastric cancer[11,12]. However in other solid malignancies, mast cells have been associated with more favorable outcomes[13–15]. The conflicting nature of these correlative findings are reminiscent of contradictory functional observations in mouse models claiming that mast cells are required for pancreatic islets tumorigenesis[16], while pancreatic adenocarcinoma occurs in a tumor microenvironment devoid of mast cells[17].

Interleukin (IL)-33 is a danger-associated signal that can serve as a molecular alarmin when released upon necroptotic and necrotic cell death including death of cancer cells[18–21]. However, cell death-independent IL-33 release can also occur[22,23]. IL-33 signals through the heterodimeric ST2 receptor, encoded by $Il1rl1$ gene, which is constitutively expressed on the surface of some innate immune cells including mast cells[24], innate lymphoid cells type 2 (ILC2)[25], and regulatory T-cells (Treg)[26,27]. IL-33/ST2 signaling is involved in triggering innate immune responses upon parasite and viral infections, and has been identified as an important mast cell activating factor[24,28] in the context of allergy[29]. Furthermore, elevated IL-33 expression was associated with poor outcomes in patients with gliomas[30], ovarian[31], as well as head and neck cancers[32]. However, predicting the outcome of IL-33/ST2 signaling in malignancies remains uncertain with both tumor promoting as well as tumor restricting activities being reported in knockout mouse models[33–36].

Here, we employ preclinically validated mouse models of gastric cancer and corresponding patient biopsies to functionally elucidate the role of mast cells during gastric tumorigenesis. Our genetic analysis reveals a linear signaling axis initiated by tumor epithelial-derived IL-33 that activates mast cells to produce a chemotactic cytokine expression signature. These factors promote the accumulation of TAMs, which in turn sustain tumor angiogenesis and growth in mice. In gastric cancer patients, this mast cell activation signature, alongside markers for tumor-associated macrophages, correlates with decreased patient survival. Our findings delineate an IL-33/mast cell/macrophage axis, which affords a clinical opportunity for the treatment of gastric cancer.

## Results

**Increased mast cell density in human intestinal-type gastric cancer and in corresponding mouse models.** In order to characterize the role of mast cells in gastric cancer, we initially investigated the mast cell frequency in $gp130^{FF}$ mice, a preclinically validated model for spontaneously occurring intestinal-type gastric cancer[37,38]. Gastric tumors in $gp130^{FF}$ mice, which harbor a knock in germline mutation in the shared IL-6/IL-11 receptor subunit gp130, arise from excessive IL-6/IL-11 dependent STAT3 activity; these tumors remain associated with chronic inflammation and immune cell infiltration. We used toluidine blue stains to quantify mast cells, and observed profound accumulation of mast cells in the submucosa of the gastric antrum in $gp130^{FF}$ mice compared to wild-type mice. Importantly, mast cells in the submucosa adjacent to gastric tumors were more abundant than the mast cells in the normal antrum-associated submucosa of $gp130^{FF}$ mice (Fig. 1a, b). In addition, mast cell numbers in $gp130^{FF}$; $Stat3^{+/-}$ compound mutant mice, which neither develop gastric tumors nor the other pan-inflammatory changes observed in their Stat3-proficient $gp130^{FF}$; $Stat3^{+/+}$ littermates[39,40], remained comparable to those in wild-type mice.

To ascertain that submucosal mast cell accumulation correlated with tumorigenesis independently of the nature of the oncogenic driver mutation, we also assessed mast cell distribution in Tg ($Tff1$-CreERT2); $Pik3ca^{H1047R/+}$; $Pten^{fl/fl}$ mice. In this model, conditional activation of a pan-glandular gastric epithelium-specific Cre-recombinase induces aberrant activation of phosphatidylinositol 3-kinase (PI3K), a pathway frequently mutated in human gastric cancer. Akin to our observations in $gp130^{FF}$ mice, mast cells also accumulated in the submucosa of tumors in Tg ($Tff1$-CreERT2); $Pik3ca^{H1047R/+}$; $Pten^{fl/fl}$ mice (Supplementary Fig. 1a, b). Strikingly, toluidine blue, alcian blue/safranin O, and May–Grunwald–Giemsa stains for granular content all consistently revealed that mast cells localized to the submucosal layers of the antrum and body of the stomach, rather than within the tumors (Supplementary Fig. 1c).

We next assessed the stromal abundance of mast cells in human biopsies collected from patients with either histopathologically classified diffuse or intestinal-type gastric cancer, or patients with preneoplastic intestinal metaplasia or chronic gastritis. Similar to our observation in mice, we observed a profound submucosal accumulation of toluidine-blue positive mast cells in intestinal-type gastric cancer. Strikingly, mast cells numbers were not elevated in the submucosa of patients with gastritis or intestinal metaplasia (Fig. 1c, d). Collectively, our results show increased mast cell abundance in the tumor-adjacent submucosa of intestinal-type gastric adenocarcinoma in human and mice, irrespective of the nature of the oncogenic driver mutation(s).

**Mast cell deficiency diminishes $gp130^{FF}$-driven tumor growth.** Having established a reproducible correlation between submucosal mast cell density and gastric cancer in humans and mice, we next determined the functional role of mast cells during tumor initiation and progression. For this we exploited the $gp130^{FF}$ model because of its high disease concordance with the human GC subtype, reproducible penetrance, onset, and progression of gastric tumorigenesis.

In order to establish whether submucosal mast cell accumulation preceded tumorigenesis, we analyzed stomach sections of 4-week-old $gp130^{FF}$ mice prior to onset of tumor formation. We detected increased mast cell density in the gastric submucosa of these tumor-free $gp130^{FF}$ mice when compared to the submucosa of age-matched WT littermates (Supplementary Fig. 2a). To functionally determine whether mast cells indeed promote gastric tumor growth, we crossed $gp130^{FF}$ mice with mast cell-deficient C57BL/6 c-Kit$^{W-sh/W-sh}$ mice carrying an inversion of the 5′-regulatory region of the $c$-$Kit$ gene that results in hypomorphic expression of the corresponding stem cell factor receptor protein.

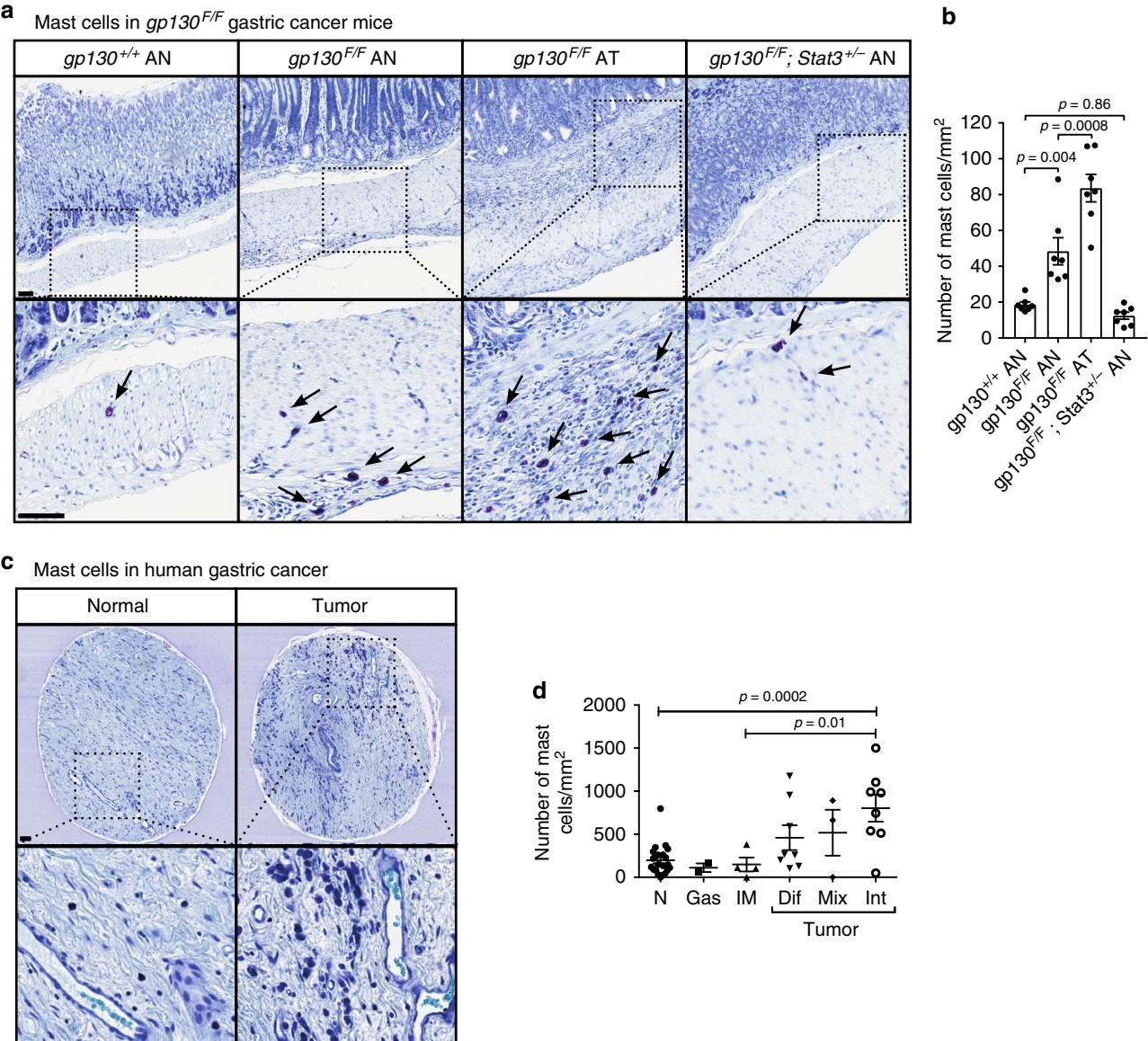

**Fig. 1** Submucosal mast cell numbers are increased in gastric cancer in mice and humans. **a** Representative cross sections of stomachs of 100 day old *gp130* mutant mice of the indicated genotype and stained with toluidine blue showing the affected antrum (AN) and antral tumor (AT), respectively. Mast cells appear purple (arrows). Scale bars = 50 µm. **b** Quantification of submucosal mast cell in sections depicted in (**a**). *n* = 7 mice per cohort obtained from two independent experiments, one-way ANOVA F (DFn = degree of freedom nominator, Dfd = degree of freedom denominator) = 34.96 (3, 24). **c** Representative sections of toluidine blue-stained biopsy cores of human gastric cancer (GC) and adjacent submucosa and of normal stomach submucosa. Scale bars = 50 µm. **d** Quantification of submucosal mast cell in sections depicted in **c**. Each symbol represents an individual patient biopsy from submucosa of normal (N; *n* = 22), gastritis (Gas; *n* = 2), intestinal metaplasia (IM; *n* = 4), diffuse (Dif; *n* = 8), mixed (Mix; *n* = 3), and intestinal-type (Int; *n* = 8) gastric cancer. One-way ANOVA F (DFn, Dfd) = 5.809 (5, 41). Data are represented as mean ± SEM, with *p* < 0.05 considered significant. Source data are provided as a Source Data file. See also related Supplementary Fig. 1

Accordingly, *c-Kit*[W-sh/W-sh] mice exhibit a substantial reduction of mast cells in all tissues[41–43], but do not suffer from anemia, sterility and lethality associated with complete c-Kit deficiency[41]. Likewise, we did not observe differences in the peripheral blood composition between *gp130*[FF] and *gp130*[FF]; *c-Kit*[W-sh/W-sh] compound mutants and we confirmed that EpCAM positive tumor cells lack c-Kit expression (Supplementary Figs. 2b, c). However, *gp130*[FF]; *c-Kit*[W-sh/W-sh] compound mice had significantly smaller and fewer tumors than their age and sex-matched *gp130*[FF] littermates (Fig. 2a–c). This reduction in tumor burden coincided with reduced tumor cell proliferation rather than

increased apoptosis, and was associated with a reduction of CD31[+] endothelial cells (Fig. 2d and Supplementary Fig. 2d).

To exclude that the anticancer effect of the *c-Kit*[W-sh] hypomorphic allele was not in part mediated by hematopoietic cells other than mast cells, we generated mast cell-deficient *gp130*[FF]; *Cpa3-Cre*; *Mcl1*[fl/fl] gastric cancer mice, where mast cell-specific carboxypeptidase A3 (Cpa3) promoter driven Cre recombinase activity leads to the deletion of *Mcl1* prosurvival gene. As a consequence *Cpa3-Cre*; *Mcl1*[fl/fl] mice retain less than 10% mast cells and have reduced numbers of basophils, while all other hematopoietic cell populations remain unaffected[44]. We

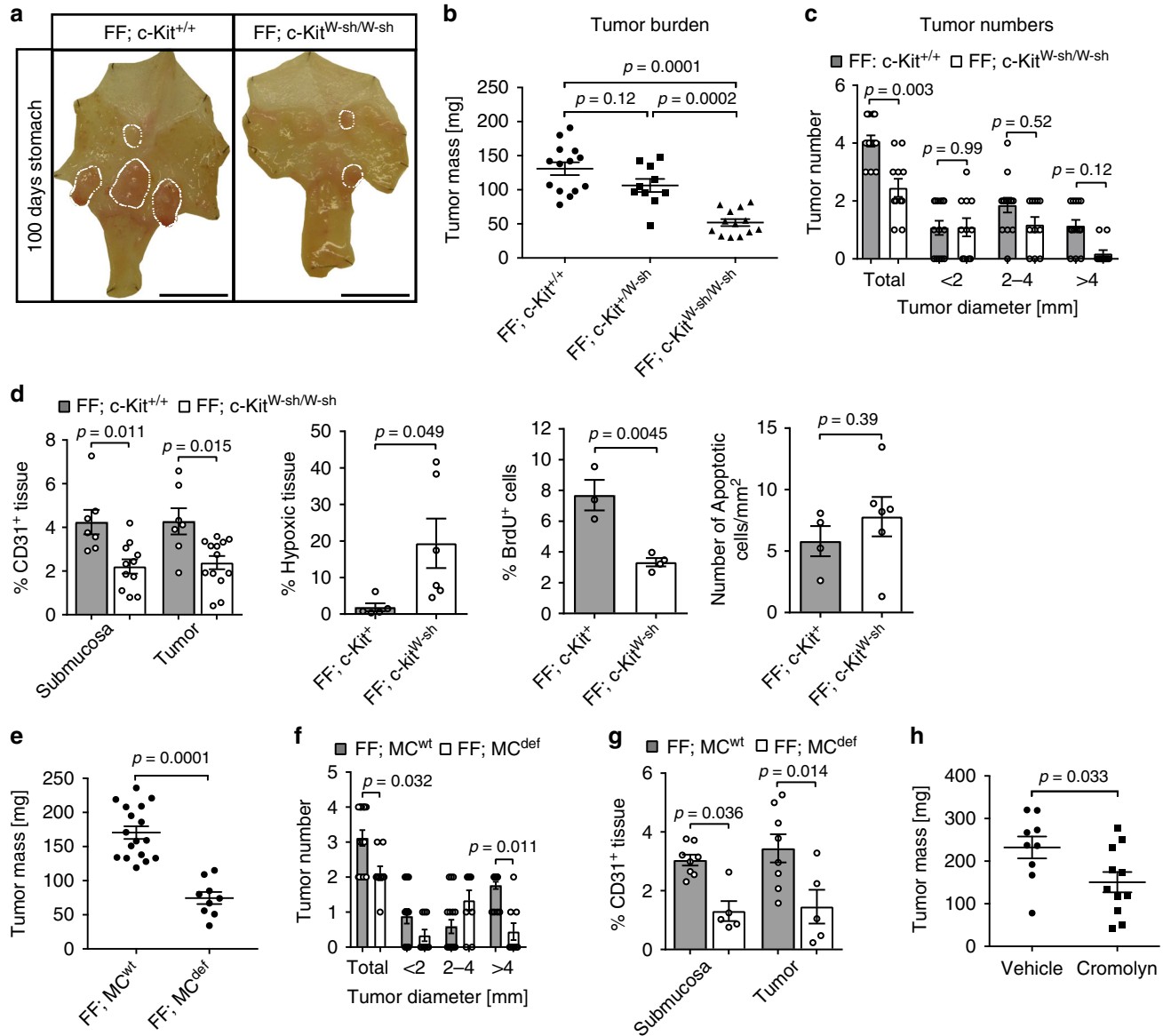

**Fig. 2** Gastric tumor burden is reduced in mast cell–deficient $gp130^{FF}$ tumor mice. **a** Representative whole mounts of pinned out stomachs, from 100-day-old $gp130^{FF}$; $c$-$Kit^{+/+}$ and mast cell-deficient $gp130^{FF}$; $c$-$Kit^{W-sh/W-sh}$ mice. Scale bars = 1 mm. **b** Quantification of total tumor burden per mouse as in (**a**). Each symbol represents an individual mouse. One-way ANOVA F (DFn, Dfd) = 25.97 (2, 24). **c** Enumeration of total tumor number and tumor size distribution as in **a** from $gp130^{FF}$; $c$-$Kit^{+/+}$ ($n = 14$) and $gp130^{FF}$; $c$-$Kit^{W-sh/W-sh}$ mice ($n = 11$). One-way ANOVA F (DFn, Dfd) = 24.59 (7, 92). **d** Quantification of CD31, Hypoxyprobe (hypoxia), BrdU (proliferation), or ApopTag (apoptosis) on immunostained gastric tumors sections. Number of mice (n) for CD31: FF; c-Kit$^{+/+}$ $n = 7$, FF; c-Kit$^{W-sh/W-sh}$ $n = 11$ with ANOVA F (DFn, Dfd) = 7.075 (3, 34); Hypoxyprobe: FF; c-Kit$^+$ $n = 5$, FF; c-Kit$^{W-sh}$ $n = 6$ with t-test + Welch correction's t (df) = 2.55 (5.256); BrdU: FF; c-Kit$^+$ $n = 3$, FF; c-Kit$^{W-sh}$ $n = 4$ with t-test's t (df) = 4.905 (5); Apoptosis: FF; c-Kit$^+$ $n = 4$, FF; c-Kit$^{W-sh}$ $n = 6$ with t-test's t (df) = 0.89 (8). **e** Quantification of total tumor burden per mouse of mast FF; MC$^{wt}$ (genotypes: FF; Cpa3-Cre$^{neg}$; Mcl1$^{fl/fl}$ and FF,Cpa-Cre;Macl1$^{+/+}$) and mast cell-deficient FF; MC$^{def}$ (genotype: FF; Cpa3-Cre; Mcl1$^{fl/fl}$) mice. t-test's t (df) = 6.72 (24). **f** Enumeration of the tumor number per mouse of FF; MC$^{wt}$ ($n = 17$ mice) and mast cell-deficient FF; MC$^{def}$ ($n = 9$ FF; Cpa3-Cre; Mcl1$^{fl/fl}$ mice. One-way ANOVA F (DFn, Dfd) = 23.25 (7, 96). **g** CD31 angiogenic staining quantification of stomachs from (**e**, **f**). FF; MC$^{wt}$ ($n = 8$ mice) and FF; MC$^{def}$ ($n = 5$) with one-way ANOVA F (DFn, Dfd) = 6.79 (3, 22). **h** Quantification of total tumor burden in $gp130^{FF}$ mice after 6 weeks administration of cromolyn (mast cell degranulation inhibitor) or vehicle. Each symbol represents an individual mouse and data was generated in three independent experiments. t-test's p value is shown and t (df) = 2.313 (18). Data are represented as mean ± SEM, with p values $p < 0.05$, being considered significant. Source data are provided as a Source Data file. See also related Supplementary Fig 2

confirmed that $gp130^{FF}$; $Cpa3$-$Cre$; $Mcl1^{fl/fl}$ mutant mice lack mast cells in their stomachs, while their $gp130^{FF}$; $Cpa3$-$Cre$; $Mcl1^{+/+}$ littermates display normal mast cell density (Supplementary Fig. 2e). Importantly, mast cell-deficient $gp130^{FF}$;$Cpa3$-$Cre$; $Mcl1^{fl/fl}$ mice had significantly reduced tumor mass and tumor number compared to their mast cell-proficient controls (Fig. 2e, f), and this observation coincided with reduced

angiogenic vessel density in the tumors of $gp130^{FF}$; $Mcl1^{fl/fl}$ mice (Fig. 2g).

To assess whether therapeutic mast cell manipulation could reduce the burden of established tumors, we exploited sodium cromoglycate (cromolyn) as a blocking agent for mast cell degranulation in patients. We treated tumor-bearing $gp130^{FF}$ mice for 6 weeks with cromolyn, which significantly decreased

tumor burden when compared to vehicle-treated mice (Supplementary Fig. 2f and Fig. 2h). Akin to our observations genetic observations in gp130$^{FF}$; c-Kit$^{W-sh/W-sh}$ and gp130$^{FF}$; Cpa3-Cre; Mcl1$^{fl/fl}$ mice, cromolyn treatment of gp130$^{FF}$ mice also reduced macrophage accumulation, tumor angiogenesis and proliferation (Supplementary Fig. 2g). These findings demonstrate the importance of mast cells and their released products for gastric tumor growth and suggest that mast cells promote tumor proliferation through mechanisms involving angiogenesis.

In the gp130$^{FF}$ mutant mice all cells harbor the gp130$^{F}$ mutation. When expressed, the gp130$^{F}$ mutant protein increases Stat3 signaling in response to IL-6 family cytokines. Because mast cells express the gp130 coreceptor and can respond to IL-6 family cytokines[45,46], we next excluded the possibility that the gp130$^{F}$ allele in mast cells may augment their tumor-promoting ability. For this, we reconstituted the hematopoietic compartment of 6-week-old lethally irradiated gp130$^{FF}$ hosts with the bone marrow from WT (gp130$^{+/+}$) or gp130$^{FF}$ mice (Supplementary Fig. 2h). Indeed, WT → gp130$^{FF}$ and gp130$^{FF}$ → gp130$^{FF}$ bone marrow chimeras showed not only similar mast cell frequencies in the gastric submucosa but also comparable tumor burden, and macrophage infiltration in the submucosa and tumor core (Supplementary Fig. 2i). These results indicate that the systemic presence of the gp130$^{FF}$ mutation did not skew hematopoietic cells towards a tumor-promoting role, nor did it increase the mast cell abundance within the gastric tumor microenvironment.

**Macrophages are reduced in gastric tumors in a mast cell-dependent manner.** Because we observed mast cells outside of the tumor cores, we surmised that tumor-promoting effect of mast cells might occur indirectly by affecting the composition and/or function of tumor infiltrating immune cells. We therefore enumerated tumor-free and tumor-containing stomach sections collected from gp130$^{FF}$ and mast cell-deficient gp130$^{FF}$; c-Kit$^{W-sh/W-sh}$ mice for the F4/80 macrophage, CD3 T lymphocyte and B220 B-lymphocyte markers. While we observed a 30% reduction in macrophage frequency in the submucosa and cores of tumors from gp130$^{FF}$; c-Kit$^{W-sh/W-sh}$ mice, the lymphocyte (CD3$^{+}$), Treg (Foxp3$^{+}$), and CD8$^{+}$ T cell frequencies remained unchanged (Fig. 3a, b). Reduced macrophage abundance within the tumor core and submucosa was also observed in the mast cell-deficient gp130$^{FF}$; Cpa3-Cre; Mcl1$^{fl/fl}$ mice (Fig. 3c) and cromolyn-treated gp130$^{FF}$ mice (Supplementary Fig. 2g). Because genetic or pharmacological interference with mast cells not only inhibited tumor growth but concomitantly also reduced the abundance of tumor-associated macrophages, we postulate that a hierarchical mast cell—macrophage relationship underpins the growth of gastric cancer.

**Macrophage infiltration coincides with gastric tumorigenesis.** To ascertain that the accumulation of tumor-associated macrophages was a common occurrence for intestinal-type gastric cancers, we assessed macrophages density in the submucosa and mucosa of tumor-bearing gp130$^{FF}$ and Tg(Tff1-CreERT2); Pik3-ca$^{H1047R/+}$; Pten$^{fl/fl}$ mice. We noted an increased abundance of macrophages in the submucosa and the mucosa of either model (Fig. 3d, e and Supplementary 3a, b), consistent with the increased abundance of submucosal mast cells in both models (Fig. 1a, b and Supplementary Fig. 1a, b). To establish whether the increased macrophage accumulation occurs prior to tumor formation, akin to our observations with mast cells, we analyzed naïve gp130$^{FF}$ mice at 4 weeks of age. In these mice macrophage numbers were increased in the submucosal layers, but remained similar between the precancerous mucosa of gp130$^{FF}$ and of WT mice (Supplementary Fig. 3c, d). Collectively, these findings positively correlate macrophage numbers with gastric tumor

growth and suggest a potential macrophage contribution to tumor formation and progression.

**Macrophages are required for gastric tumorigenesis in gp130$^{FF}$ mice.** Macrophages have been established as drivers of tumor growth and cancer therapy targets for several tumor entities[47,48]. To formally investigate a functional contribution of macrophages to gastric tumorigenesis, we generated macrophage-deficient gp130$^{FF}$; Csf1r$^{-/-}$ mice, because expression of the Csf1r gene and its corresponding receptor protein for macrophage colony-stimulating factor is required for maturation of bone marrow-derived macrophages (BMDM) from their monocytic precursors. Strikingly, the few surviving gp130$^{FF}$; Csf1r$^{-/-}$ mice remained completely free of gastric tumors (Fig. 4a, b). The submucosal mast cell density was not altered in the macrophage-deficient animals (Fig. 4c). We next confirmed that this phenotype reversion was not related to the runted phenotype of gp130$^{FF}$; Csf1r$^{-/-}$ mice associated with their feeding requirement for mashed chow to overcome the Csf1R deficiency-dependent absence of teeth. We therefore depleted macrophages in adult tumor-bearing gp130$^{FF}$ mice by systemic administration of clodronate-loaded liposomes (clodrosomes) for 6 consecutive weeks and observed a significant reduction of tumor burden when compared to the vehicle control cohort (Fig. 4d). Although toluidine-staining confirmed that clodrosome treatment did not reduce the abundance of submucosal mast cells, gastric tumors and adjacent submucosa from the clodrosome cohort revealed reduced density of CD31-positive microvessels (Fig. 4e and Supplementary Fig. 4a).

The reduced tumor-burden and microvessel density observed in the clodronate cohort were similar to our findings from the mast cell depletion experiments. We next aimed to exclude that they relate to off-target-effects associated with bisphosphonate (i.e., clodronate) administration. We therefore treated tumor-bearing gp130$^{FF}$ mice with the small molecule Csfr1/c-kit/Flt3 kinase inhibitor PLX3397 (Pexidartinib). We supplied PLX3397 in the food to tumor-bearing gp130$^{FF}$ mice for 4 weeks (acute therapeutic) and continued a cohort for a further 4 weeks without PLX3397 treatment (follow-up) (Supplementary Fig. 4b). Acute PLX3397 treatment revealed a significant reduction in tumor burden, although this antitumoral effect was reversible as the tumor burden increased once PLX3397 treatment ceased in the follow-up cohort (Fig. 5a). In the acute treatment cohort, we confirmed by immunohistochemical staining a significant reduction of F4/80-positive macrophages accompanied by reduced density of CD31-positive angiogenic vessels and increased staining for tumor-hypoxia (Fig. 5b and Supplementary Fig. 4c). However, owing to the specificity of PLX3397, we also observed a decrease of tumor-adjacent submucosal mast cells. All these parameters reverted back to their pretreatment state following a drug-free period in the follow-up cohort (Supplementary Fig. 4c).

Based on our proposed role for macrophages to mediate the tumor promoting effect of mast cells, we hypothesized that the tumor-associated macrophages in the gastric lesions of gp130$^{FF}$ mice should have a tumor-promoting alternatively activated endotype. We therefore compared expression of macrophage endotype markers between naïve BMDM and tumor-associated macrophages, and used standard in vitro polarization protocols for BMDM to generate a reference population for alternative activation. For the tumor-associated macrophage populations, this analysis revealed high expression of the alternative activation markers Arg1, Fizz1, and Mrc1, which coincided with reduced mRNA levels of the classically activated marker Nos2 (Fig. 5c). Consistent with the pronounced presence of microvasculature in tumors of these gp130$^{FF}$ mice, we also recorded elevated expression of angiogenic Vegfa in the tumor-associated

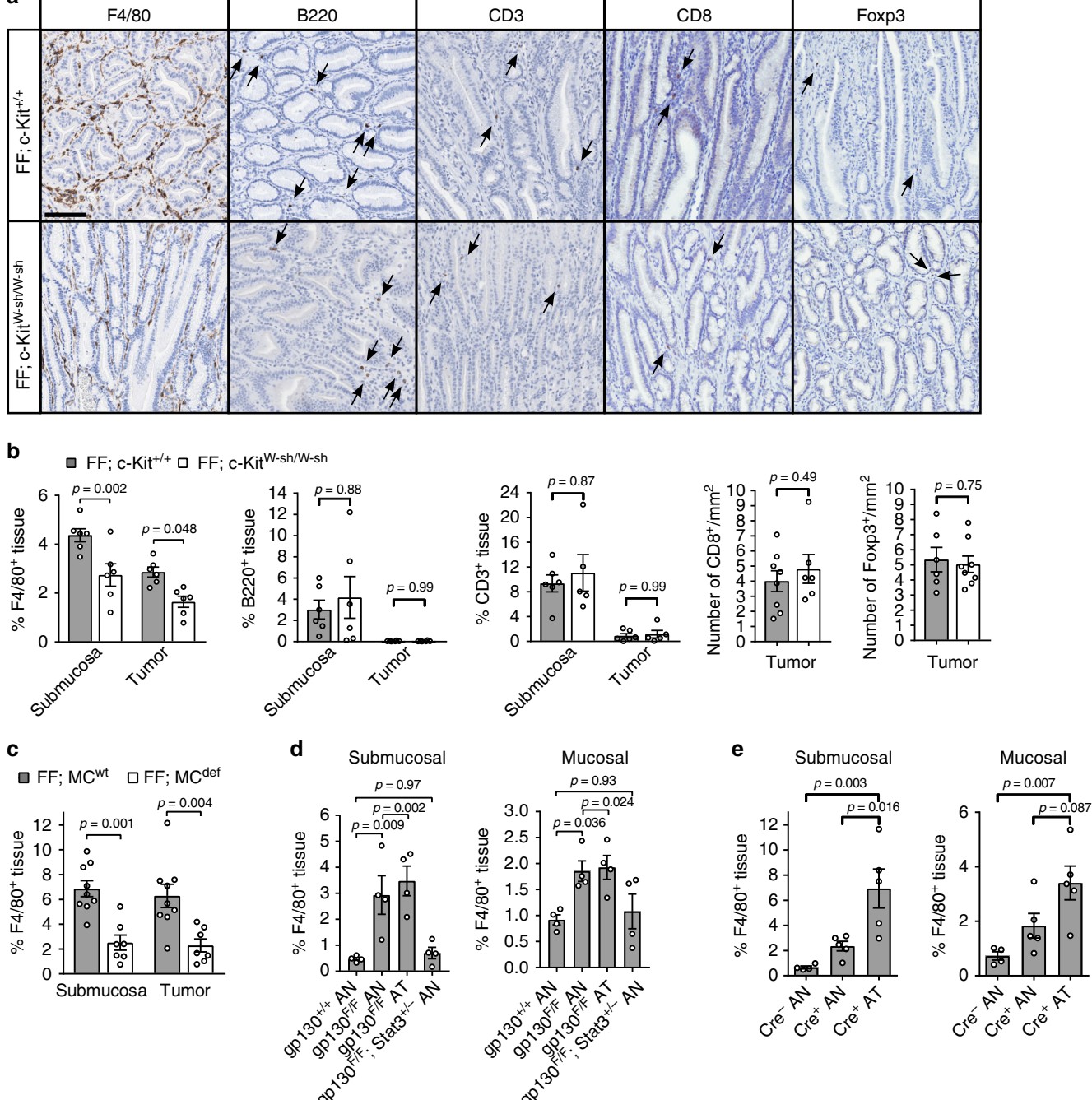

**Fig. 3** Mast cell depletion reduces macrophage infiltration in gastric tumors of *gp130*FF mice. **a** Representative images of tissue sections immunostained for macrophages (F4/80), B cells (B220) or T cells (CD3, CD8, Foxp3) in gastric tumor sections of *gp130*FF; *c-Kit*+/+ and mast cell-deficient *gp130*FF; *c-Kit*W-sh/W-sh mice. Arrows indicate specific positive cell staining; scale bars = 100 μm. **b** Quantification of F4/80, B220, or CD3 expressing cells in tumor and tumor-associated submucosal layers of sections from (**a**). F4/80: *n* = 6 each group, one-way ANOVA F (DFn, Dfd) = 13.43 (3, 20); B220: *n* = 6 each group, one-way ANOVA F (DFn, Dfd) = 3.782 (3, 20); CD3: *FF*; *c-Kit*+/+ *n* = 6, *FF*; *c-Kit*W-sh/W-sh *n* = 5, one-way ANOVA F (DFn, Dfd) = 11.55 (3, 18); CD8: FF, c-Kit+/+ *n* = 8, *FF*; *c-Kit*W-sh/W-sh *n* = 6, *t*-test t (df) = 0.71 (12); Foxp3: FF, c-Kit+/+ *n* = 6, *FF*; *c-Kit*W-sh/W-sh *n* = 8, *t*-test t (df) = 0.33 (12). Data from two independent experiments were analyzed. **c** F4/80+ cells frequency in stomach submucosa and tumors of *FF*; *MC*wt mice (*n* = 9) and mast cell-deficient *FF*; *MC*def mice (*n* = 7 *FF;Cpa3-Cre; Mcl1*fl/fl). One-way ANOVA F (DFn, Dfd) = 10.81 (3, 28). **d** Quantification of F4/80+ cells in unaffected antrum (AN) or antrum tumor (AT) in submucosa or mucosa of mice of the indicated genotype. All groups *n* = 4. Submucosal: one-way ANOVA F (DFn, Dfd) = 10.12 (3, 12); Mucosal: one-way ANOVA F (DFn, Dfd) = 5.086 (3, 12). **e** Quantification of F4/80+ cells in unaffected antrum (AN) or antrum tumors (AT) in submucosa or mucosa of mice from the Tg(*Tff1-CreERT2*); *Pik3ca*H1047R/+; *Pten*fl/fl strain that either harbor (Cre+) or lack (Cre−) the *Tff1-CreERT2* driver. *Cre*− AN *n* = 4, *Cre*+ AN and *Cre*+ AT *n* = 5 for both. Analyses from two independent experiments. Submucosal: one-way ANOVA F (DFn, Dfd) = 10.52 (2, 11); Mucosal: one-way ANOVA F (DFn, Dfd) = 7.485 (2, 11). Data are represented as mean ± SEM, with *p* values *p* < 0.05 considered being significant. Source data are provided as a Source Data file. See also related Supplementary Fig. 3

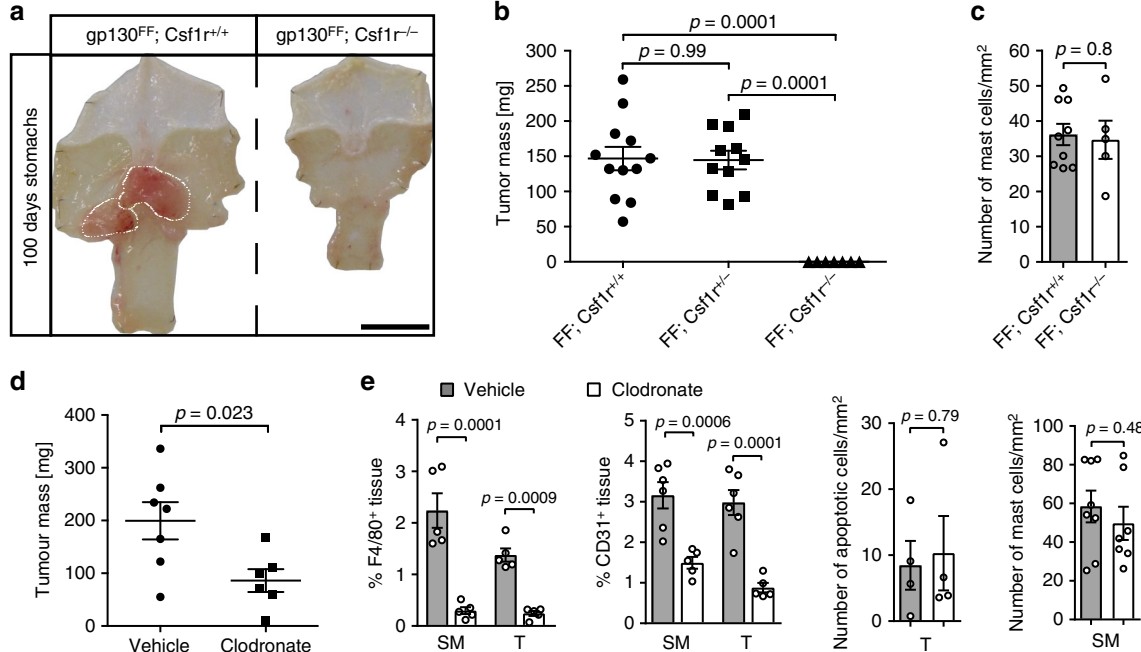

**Fig. 4** Tumor burden in *gp130*[FF] mice diminishes upon macrophage depletion. **a** Representative whole mount of stomachs from 100-day-old mice of the indicated genotype. Scale bar = 1 mm. **b** Quantification of total tumor burden per mouse as in (**a**). Each symbol represents an individual mouse. Due to ill health some *gp130*[F/F]; *Csf1r*[−/−] mice had to be analyzed before the 100 day endpoint; average mouse age per groups were: *gp130*[F/F]; *Csf1r*[+/+] 88.2 days, *gp130*[F/F]; *Csf1r*[+/−] 90.4 days, and *gp130*[F/F]; *Csf1r*[−/−] 81.6 days. One-way ANOVA F (DFn, Dfd) = 27.55 (2, 27). **c** Quantification of mast cell density in antral submucosa of indicated genotype was performed with *gp130*[F/F]; *Csf1r*[+/+] $n = 9$ and *gp130*[F/F]; *Csf1r*[−/−] $n = 5$ biological samples from two independent experiments $t$-test t (df) = 0.262 (12). **d** Assessment of total tumor burden in *gp130*[FF] mice after 6 weeks administration of clodronate (50 μl of a clodrosome solution containing 5 mg/ml clodronate) or vehicle. Each symbol represents an individual mouse; data from two independent experiments ($t$-test t (df) = 2.622 (11)). **e** Quantification of F4/80, CD31, ApopTag, or toluidine blue (for detection of mast cells) stained sections of gastric tumors (T) and tumor-associated submucosa (SM) from *gp130*[FF] mice of the indicated treatment cohort. F4/80: all $n = 5$, one-way ANOVA F (DFn, Dfd) = 27.17 (3, 16); CD31: $n = 6$ (SM tissues), $n = 5$ (T tissues), one-way ANOVA F (DFn, Dfd) = 18.15 (3, 18); apoptotic cells: $n = 4$ (both groups), $t$-test t (df) = 0.27 (6); mast cells: $n = 8$ (vehicle) and $n = 7$ (Clodronate), $t$-test t (df) = 0.73 (13). Data are represented as mean ± SEM, with $p$ values $p < 0.05$ considered being significant. Source data are provided as a Source Data file. See also related Supplementary Fig. 4

macrophage population when compared to the naïve BMDM population from the same mice (Fig. 5c). Given the alternative activated endotype of tumor-associated macrophages, our data collectively suggests that their abundance is likely to be a rate-limiting factor for the establishment of tumor microvasculature and the growth of gastric tumors.

**Tumor-derived IL-33 activates mast cell secretion and expression of macrophage-attracting chemokines.** Although our observations thus far suggested a functional mast cell–macrophage relationship, it remained to determine the molecular network underpinning this hierarchy including how its activation could occur. We therefore focused on IL-33 as a potential tumor-derived activator of mast cells, because IL-33 was reported to serve as an alarmin that is generated and released by necrotic cells, and that triggers mast cell degranulation[49]. Indeed, we detected increased expression of *Il33* in the EpCam[+]/CD45[−] epithelial components of the tumors in *gp130*[FF] mice with a reciprocal expression of the *Il1rl1* gene in the hematopoietic CD45[+]/EpCam[−] compartment of the antrum, which encodes the cognate St2 receptor subunit for IL-33 (Fig. 6a). In line with the reported storage of IL-33 protein in vesicles, we detected increased levels of *Il33* mRNA and IL-33 protein in whole tumor tissue (Supplementary Figs. 5a, b). Immunofluorescence staining revealed that IL-33 protein localized most prominently to the peripheral regions of the epithelial components of the tumors (Fig. 6b) coinciding with the location of the highest level of cellular turn-over[37].

To assess whether tumor-associated mast cells could respond to IL-33, we FACS purified the c-Kit[+]/F$_c$εR1[+] mast cells from the submucosal antrum of WT and *gp130*[FF] mice using stomachs from *gp130*[FF]; *c-Kit*[W-sh/W-sh] mice as a specificity control for our purification strategy (Supplementary Fig. 5c). We confirmed the homogeneity of the c-Kit[+]/F$_c$εR1[+] mast cell population by toluidine blue staining (Supplementary Fig. 5d) and showed prominent expression of the St2 receptor in the c-Kit[+]/F$_c$εR1[+] mast cells isolated from the stomach of *gp130*[FF] mice (Supplementary Fig. 5e). We then stimulated this cell population for 3 h with IL-33 and detected by the multiplex cytokine assay increased amounts of IL-6 and IL-13 and to lesser extent of IL-1α, IL-4, alongside elevated secretion of the macrophage growth and attracting factors Gm-csf, Mcp1, Mip-1α, and Mip-1β (Supplementary Fig. 5f and Fig. 6c). Likewise, qPCR analysis of the IL-33 stimulated mast cells also revealed increased expression of *Ccl2* (encoding Mcp1 protein), *Ccl3* (encoding Mip-1α protein), and *Ccl7* (Fig. 6d), which collectively exert chemotactic activity towards macrophages and neutrophils[50–52]. Also, IL-33-stimulation induced elevated expression of angiogenic growth factor gene *Vegfa* (Fig. 6d). In order to confirm the chemotactic activity of the mast cell produced factors on *gp130*[FF] macrophages, we performed migration assays with BMDM. Indeed, Mcp1 and Mip-1α alone induced migration significantly (Supplementary Fig. 5g).

Because *gp130*[FF]-driven gastric tumorigenesis has an absolute requirement for IL-11 signaling[38], we predicted that IL-11 could stimulate IL-33 expression in tumor-derived epithelium. To test

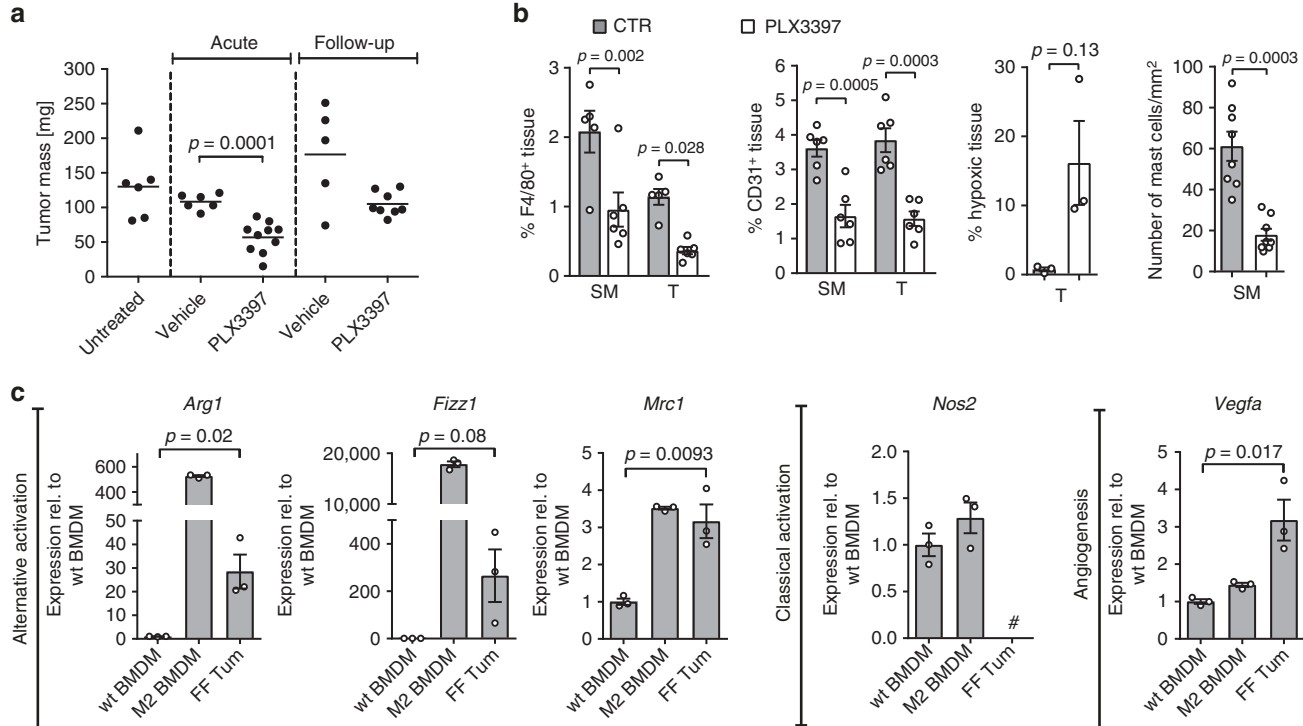

**Fig. 5** Pharmacological macrophage targeting and macrophage polarization in *gp130*[FF] tumors. **a** Total tumor burden in *gp130*[FF] mice was quantified either acutely after a 4-week-treatment period with PLX3397 (Csf1r/c-Kit/Flt3 tyrosine kinase receptor inhibitor) or vehicle or after a 4 weeks treatment-free follow-up period. Each symbol represents an individual mouse. Data from three independent experiments are presented. *t*-test t (df) = 5.23 (14). **b** Quantification of F4/80, CD31, Hypoxyprobe or toluidine-blue-stained sections of submucosa (SM) or gastric tumors (T) of *gp130*[FF] mice of the indicated acute treatment cohort. F4/80: *n* = 5 (Vehicles) and *n* = 6 (PLX3397), one-way ANOVA F (DFn, Dfd) = 12.34 (3, 18); CD31: *n* = 6, one-way ANOVA F (DFn, Dfd) = 18.33 (3,20); hypoxic tissue: *n* = 3, *t*-test with Welch's correction t (df) = 2.53 (2); mast cells: *n* = 8 (both groups), and *t*-test with Welch's correction t (df) = 5.59 (9.27). **c** qPCR expression analysis of genes associated with alternative activation (*Arg1, Fizz1, Mrc1*), classical activated (*Nos2*) and angiogenesis (*Vegfa*) of purified F4/80[+] tumor-associated macrophages (FF Tum) or following stimulation of bone marrow-derived macrophages (BMDM) from wild-type mice and stimulated either with vehicle (wt BMDM) or with IL-4/IL-13 (20 ng/ml) to induce an alternative activated endotype (M2 BMDM). *n* = 3 mice. Hash indicates that expression was below detection limit. Arg1: *t*-test t (df) = 3.8 (4); Fizz1: *t*-test t (df) = 2.39 (4); Mrc1: *t*-test t (df) = 4.69 (4); Vegfa: *t*-test t (df) = 3.96 (4). Data are represented as mean ± SEM, with *p* values *p* < 0.05 considered being significant. Source data are provided as a Source Data file. See also related Supplementary Fig. 4

this, we exposed epithelial organoids grown from *gp130*[FF]-tumors to recombinant IL-11 in vitro. While organoid growth and morphology were not altered (Fig. 6e and Supplementary Fig. 5h), we observed a marked increase in *Il33* expression upon IL-11 stimulation, which was comparable to the extent of induction of the *bona fide* IL-11/Stat3-target gene *Socs3* (Fig. 6f). Additionally, we conducted a chromatin immunoprecipitation (ChIP) experiment on DNA from gastric tumors of *gp130*[FF] mice 60 min after acute systemic administration of IL-11. The ChIP experiment revealed a Stat3-specific binding peak on chromosome 19, which maps to the 5′-region of the *Il33* gene and which contains a Stat3-binding site motif (Supplementary Fig. 5i, j). Collectively, these data indicate that tumor-derived IL-11 induces *Il33* gene expression in gastric tumor cells, which triggers the secretion as well as the de novo synthesis of cytokines and chemokines by mast cells to produce a tumorigenic immune environment.

**IL-33/St2 signaling deficiency decreases *gp130*[FF]-mediated gastric tumor growth.** To validate a functional contribution of IL-33 signaling as a master regulator of mast cell activation in the gastric tumor microenvironment, we generated IL-33 signaling deficient *gp130*[FF]; *St2*[−/−] mice. Compared to their *St2*[+/+] and *St2*[+/−]-proficient compound mutant littermates, *gp130*[FF]; *St2*[−/−] mice harbored a significantly lower overall tumor burden at 100 days of age (Fig. 7a), which arose from reduced tumor growth

rather than reduced tumor incidence (Fig. 7b). These observations coincided with a decrease in mast cell numbers in the tumor-adjacent submucosal layers and reduced density of F4/80[+] macrophages in the tumors of *gp130*[FF]; *St2*[−/−] mice as well as reduced abundance of CD31-positive microvessels (Fig. 7c).

We next confirmed that c-Kit[+]/FcεR1[+] mast cells isolated from *gp130*[FF]; *St2*[−/−] mice showed significantly decreased expression of the macrophage recruiting and inflammatory mediators *Csf2* (encoding Gm-csf), *Ccl3*, and *Il6* (Fig. 7d). Because IL-33 signaling is involved in expansion and activation of ILC2 and Tregs, which could also afford a mechanism for IL-33 to promote gastric tumorigenesis in *gp130*[FF] mice, we assessed the relative abundance of ILC2 and Tregs among all cells harboring the pan-hematopoietic surface marker CD45. We observed comparable frequency of these cells between unaffected antrum and tumors of *gp130*[FF] and *gp130*[FF]; *St2*[−/−] mice. Moreover, less than 30% of all tumor-associated ILC2 cells, and less than 15% of all Tregs, expressed the St2 receptor (Fig. 7e, f).

To investigate the involvement of mast cells in the antitumoral effect of St2-deficiency, we established bone marrow-derived mast cells (BMMC) from *gp130*[FF]; *ST2*[−/−] and *gp130*[FF]; *St2*[+/+] mice, confirmed their purity (Supplementary Figs. 6a–c) and performed adaptive transfer of BMMC into *gp130*[FF]; *St2*[−/−] hosts (Supplementary Fig. 6d). Reconstitution with St2[+] wild-type BMMC (*gp130*[FF]; *St2*[+/+]) increased tumor burden compared to littermates injected with *gp130*[FF]; *St2*[−/−] BMMCs (Fig. 7g and Supplementary

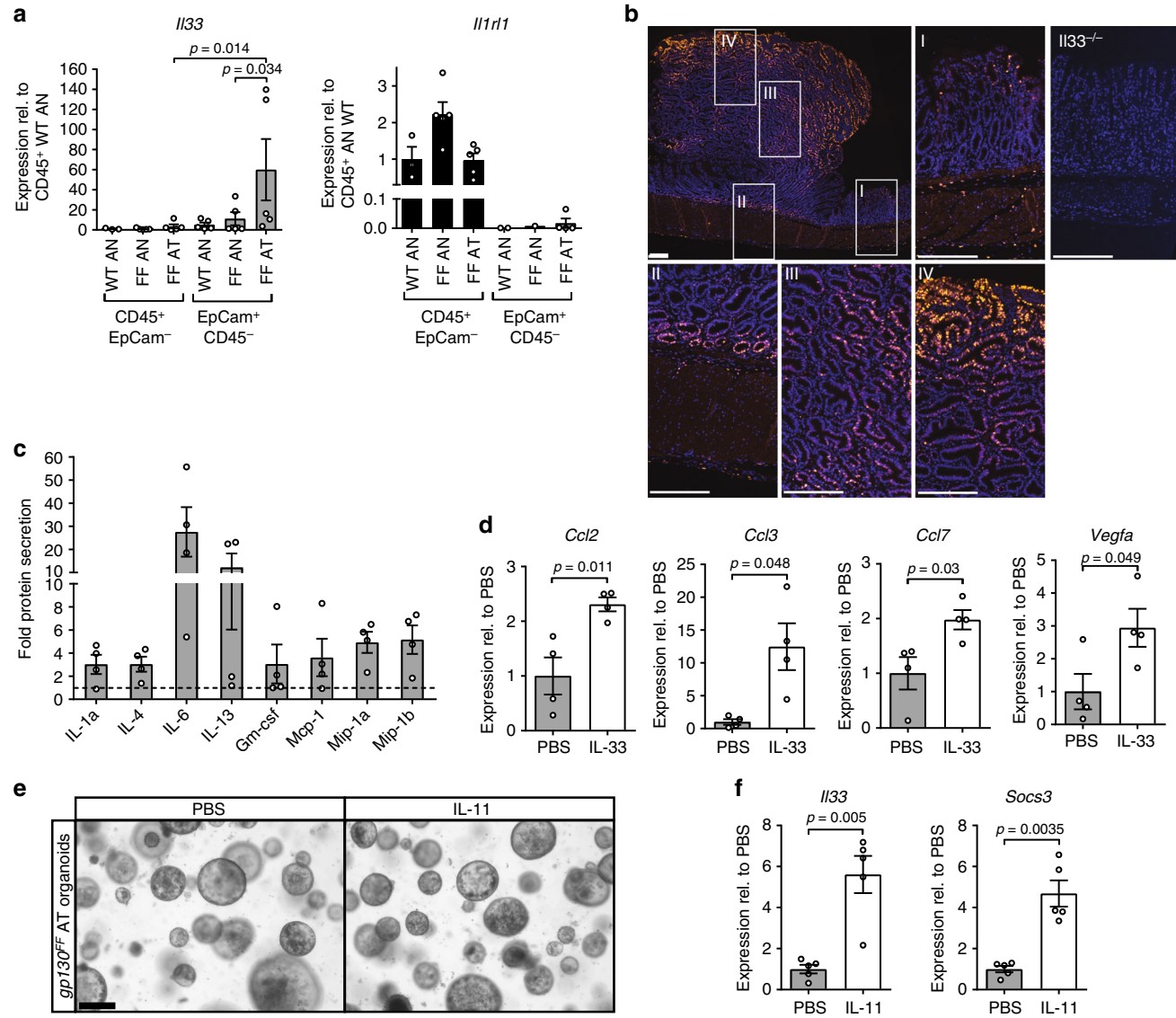

**Fig. 6** IL-33 expression in gastric tumors of $gp130^{FF}$ mice and mast cell activation analysis. **a** qPCR expression analysis of $Il33$ and full-length St2 ($Il1rl1$) genes associated with hematopoietic (CD45$^+$; EpCam$^-$) and epithelial (EpCam$^+$; CD45$^-$) cells purified from unaffected antrum (AN) or antrum tumors (AT) from wild-type and $gp130^{FF}$ (FF) mice. Data are normalized to $Gapdh$ and plotted as relative expression to CD45$^+$; EpCAM$^-$ WT AN expression. $n = 5$ mice. Il1rl1 expression in EpCam$^+$; CD45$^-$ cells was for several samples below detection limit. Data was pooled from two independent experiments. For IL33 data, one-way ANOVA was performed with F (DFn, Dfd) = 2.871(5, 22); **b** Immunofluorescence staining for IL-33 in stomachs of tumor-bearing $gp130^{FF}$ mice with insets referring to unaffected antrum (I), submucosal-tumor junction (II), tumor core (III), and tumor edge (IV). Stomachs from $Il33^{-/-}$ mice were used for specificity controls. Scale bars = 200 μm. **c** Multiplex cytokine analysis of supernatant of FACS-purified tumor-associated mast cells stimulated with IL-33 (30 ng/ml) for 3 h. Data are shown only for factors with > 3 fold increase relative to unstimulated control cultures. $n = 4$ from four independent experiments. **d** qPCR gene expression analysis in FACS-purified tumor-associated mast cells stimulated with IL-33 (30 ng/ml) for 3 h or vehicle. $n = 4$ from four independent experiments. $Ccl2$: $t$-test t (df) = 3.61 (6); $Ccl3$: $t$-test t (df) = 3.2 (3.09); $Ccl7$: $t$-test t (df) = 2.84 (6); $Vegfa$: $t$-test t (df) = 2.45 (6). **e** Representative images of organoids derived from antral tumors (AT) of $gp130^{FF}$ mice stimulated either with PBS or IL-11 (100 ng/ml for 2 days). Scale bar = 200 μm. **f** $Il33$ gene expression analysis of $gp130^{FF}$ tumor-derived epithelial organoids either stimulated with IL-11 (100 ng/ml) or with PBS for 4 h. For comparison expression of Stat3-target gene, $Socs3$, was also analyzed. $n = 5$ from two independent experiments. $Stat3$: $t$-test t (df) = 4.96 (4.42); $Socs3$: $t$-test t (df) = 5.6 (4.48). Data are represented as mean ± SEM, with $p$ values $p < 0.05$ considered being significant. Source data are provided as a Source Data file. See also related Supplementary Fig. 5

Fig. 6e). At the time point of tumor analysis 8% of the gastric submucosal mast cells stained positive for St2 expression and therefore represent the transplanted BMMC mast cells, which homed to the gastric tumor site and compete with the endogenous ST2$^{-/-}$ mast cell compartment (Supplementary Fig. 6f).

These observations further support a mechanism by which IL-33 promotes gastric tumorigenesis through mast cell activation rather than by promoting ILC2 or Treg accumulation.

**IL-33/mast cells-activity gene expression signature predicts poor survival for human intestinal-type gastric cancer.** In order to translate our preclinical findings in mice to a relevant setting in humans, we hypothesized that the extent of mast cell activation might provide retrospective prediction of patient survival. Based on the mast cell activation gene expression signature derived from our in vitro mast cell stimulation with IL-33, we analyzed a publically available data set (DErrico Gastric data set; GEO:

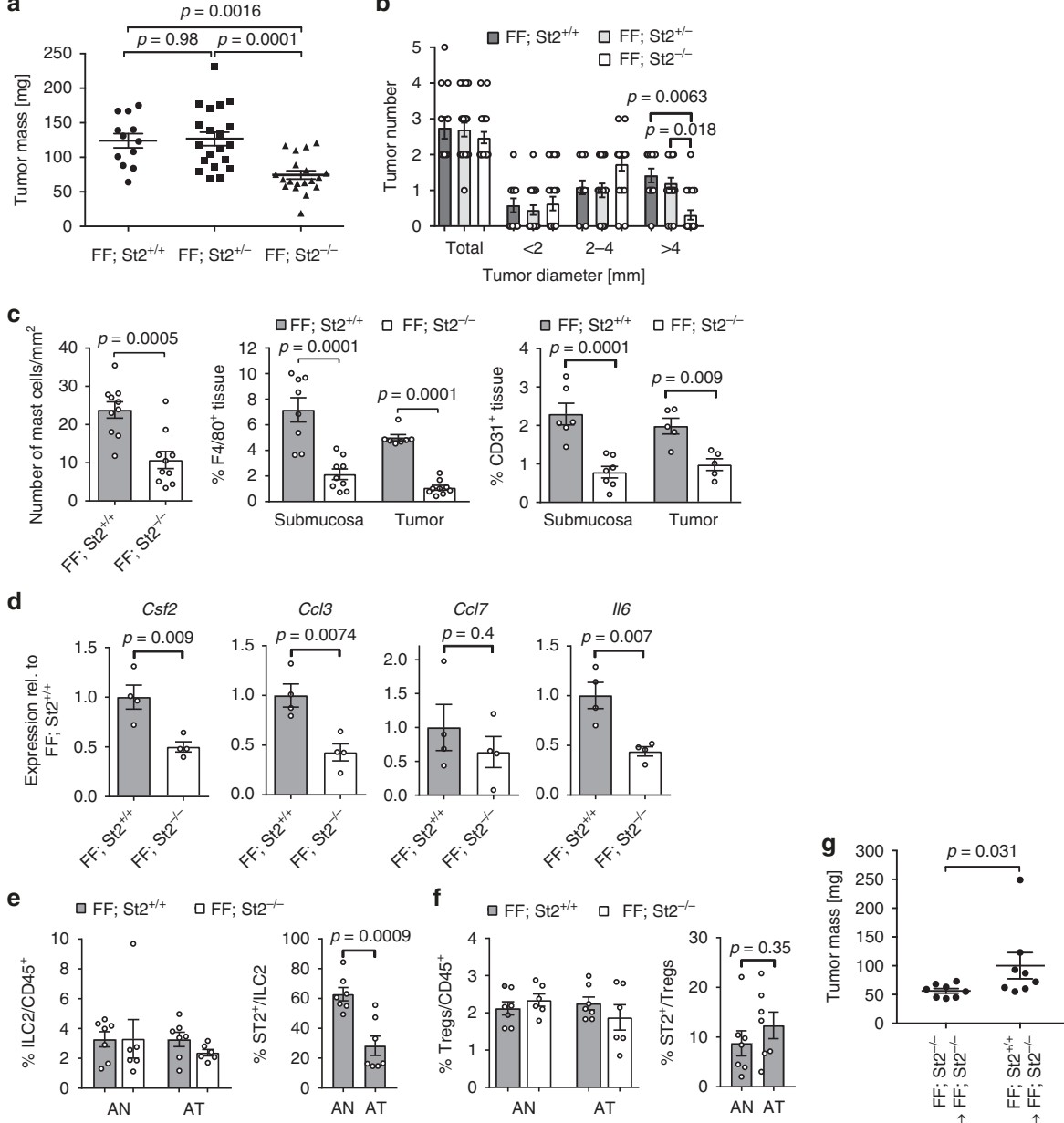

**Fig. 7** Tumor burden is reduced in St2 receptor-deficient $gp130^{FF}$ mice. **a** Quantification of total tumor burden in 100-day-old mice of the indicated genotype. Each symbol represents an individual mouse. One-way ANOVA was performed with F (DFn, Dfd) = 11.83 (2, 48). **b** Enumeration of total tumor number from mice in **a**, and of tumors following classification according to their size. $n = 12$ (FF, $St2^{+/+}$), $n = 20$ (FF, $St2^{+/-}$), and $n = 19$ (FF, $St2^{-/-}$) mice. One-way ANOVA was performed with F (DFn, Dfd) = 22.79 (11, 192). **c** Quantification of toluidine blue (for detection of mast cells; submucosal tissue), F4/80 and CD31 stained sections of gastric tumors of mice of the indicated genotype. Mast cells: $n = 10$ mice, t-test t (df) = 4.25 (18); F4/80: $n = 8$ (FF; $St2^{+/+}$), $n = 9$ (FF; $St2^{-/-}$), one-way ANOVA F (DFn, Dfd) = 27.52 (3,29); CD31: $n = 6$ (Submucosa) $n = 5$ (Tumor), and one-way ANOVA F (DFn, Dfd) = 13.6 (3,19). **d** qPCR expression analysis of chemokines expressed by FACS-purified tumor-associated mast cells from stomachs of either *FF; St2*$^{+/+}$ or *FF; St2*$^{-/-}$ mice. All $n = 4$ from two independent experiments. *Csf2*: t-test t (df) = 3.81 (6); *Ccl3*: t-test t (df) = 3.97 (6); *Ccl7*: t-test t (df) = 0.88(6); *Il6*:: t-test t (df) = 4.02 (6); **e**, **f** Flow cytometric analysis of unaffected antrum (AN) and antrum tumors (AT) of indicated genotype for the frequency of ILC2 cells (lineage⁻, Cd11b⁻, Gata3⁺), Tregs (Foxp3⁺, CD4⁺), and proportion of St2⁺ cells within these cell types. *FF; St2*$^{+/+}$ $n = 7$ and *FF; St2*$^{-/-}$ $n = 6$, from two independent experiments. ST2⁺/ILC2: t-test t (df) = 4.39 (12); ST2⁺/Treg: t-test t (df) = 0.98 (12). **g** Enumeration of total tumor burden at 14 weeks of age of *FF; St2*$^{-/-}$ host mice, which received tail vein injections of either *FF; St2*$^{-/-}$ or *FF, St2*$^{+/+}$ bone marrow-derived mast cells (BMMC) ($n = 8$ mice per group). Mann–Whitney test was performed with Mann–Whitney $U = 11.5$. Data are represented as mean ± SEM, with p values $p < 0.05$ considered being significant. Source data are provided as a Source Data file. See also related Supplementary Fig. 6

GSE13911) for differential expression of the mast cell signature comprising *CCL2, CCL3, CCL4, IL1a, IL4, IL6, IL13, CSF2*, and *CCL7*. With the exception of IL-4 and IL-13, we found that all other factors were at least 1.6-fold increase in the cancer samples ($p \leq 0.05$) (Fig. 8a). We therefore assigned the differentially expressed factors (i.e., *CCL2, CCL3, CCL4, IL1a, IL6, CSF2*, and *CCL7*) to an IL-33/mast cell activation gene expression signature for a Kaplan–Meier survival analysis (KMplot.com)[53].

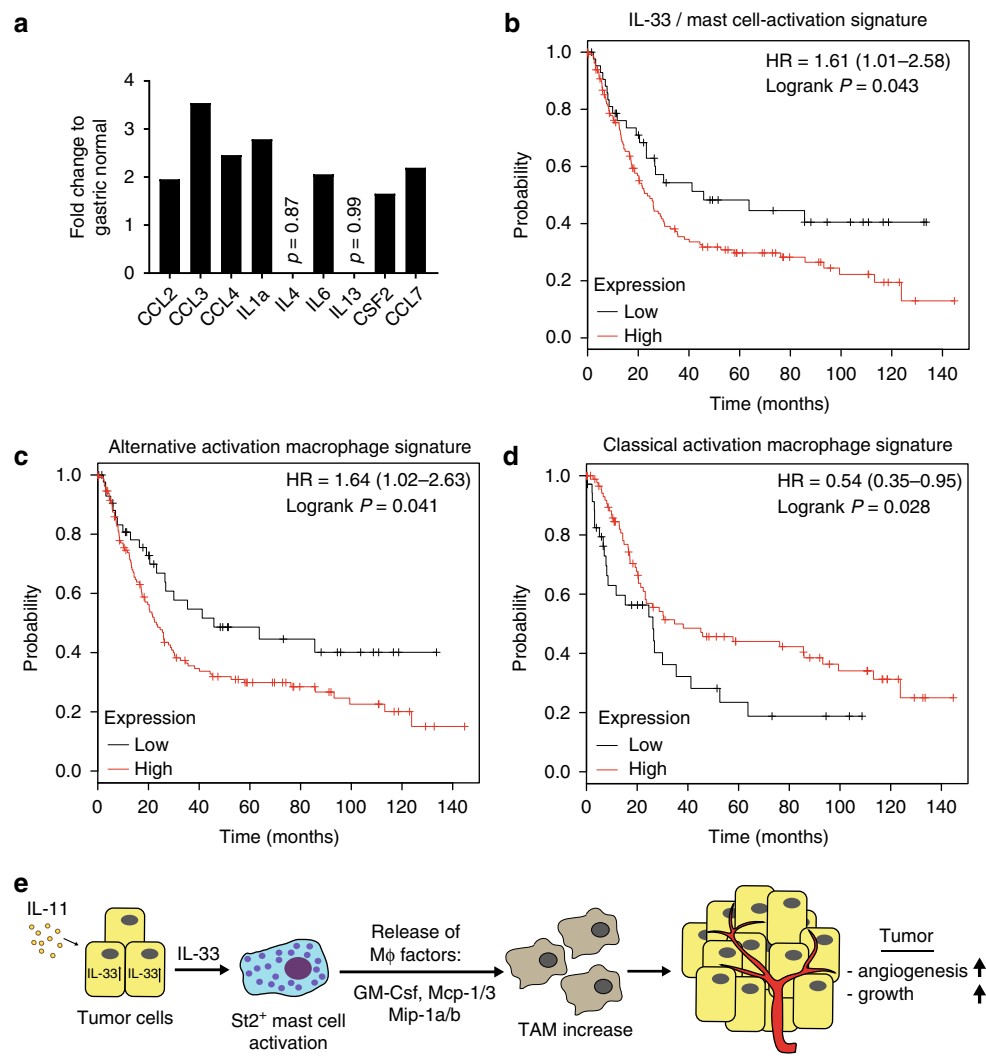

**Fig. 8** Kaplan–Meier analysis for an IL-33 - mast cell activation gene expression signature. **a** Expression analysis of genes associated with IL-33 dependent mast cell activation derived from the DERRICO gastric data set (GEO: GSE13911, extracted from oncomine.org) and comparing normal gastric mucosa ($n = 31$) and gastric intestinal-type adenocarcinoma ($n = 29$). The $p$-value for the genes is $p < 0.05$ except where indicated. **b–d** Kaplan–Meyer survival analysis was performed for human intestinal-type gastric cancer (iGC) with IL-33/ mast cell activation signature consisting of *CCL2*, *CCL3*, *CCL4*, *IL1a*, *IL6*, *CSF2*, and *CCL7* (**b**), with alternative activation macrophage expression signature (*CD163*, *CD204*, *MARCO*, *ARG1*) (**c**) and classical activation macrophage expression signature (*NOS2A*, *HLA-DRA*, *CD80*, *CD86*, *CD169*) (**d**). **e** Schematic illustration of the proposed gastric cancer growth promoting IL-11/IL-33/mast cell/ TAM signaling axis. Source data are provided as a Source Data file. See also related Supplementary Fig. 7

Remarkably, we found a lower overall survival for intestinal-type gastric cancer patients with high expression of the entire signature (HR = 1.61; logRank $P = 0.043$) (Fig. 8b). Similar to our IL-33/mast cell activation gene expression signature, a trend for better survival probability was found for high expression of a classical mast cell marker gene signature consisting of *KIT*, *FCER1G*, and *HDC* (encoding Histidine Decarboxylase) (HR = 1.42; logRank $P = 0.067$) (Supplementary Fig. 7a) as well as for ST2-receptor encoding gene IL1RL1 (HR = 1.57; logRank $P = 0.049$) (Supplementary Fig. 7b). In accordance with our mast cell–macrophage hierarchy model, Kaplan–Meier survival analysis for macrophages revealed a survival disadvantage for intestinal-type gastric cancer patients with high expression of an alternatively activated macrophage gene signature (Fig. 8c). On the other hand, high expression of a classically activated gene signature correlates with a higher survival probability (Fig. 8d). Our preclinical data argue strongly for a tumor promoting role of a signaling axis emanating from IL-11-induced, tumor cell-

derived IL-33 and the subsequent hierarchical activation cascade of mast cells and tumor-associated macrophages. Collectively, these innate immune cells enable the establishment of the microvasculature required for gastric tumor growth and may provide therapeutically actionable targets (Fig. 8e).

## Discussion

Although conflicting data indicate a dichotomous role of mast cells to either restrict or support tumorigenesis, compelling evidence implicates mast cells in promoting progression and metastasis in many solid malignancies. Retrospective studies on human gastric cancers have suggested an involvement of tumor-adjacent submucosal mast cells to late stage disease[54] and metastasis formation[12]. However, the molecular mechanism by which submucosal mast cells contribute to gastric tumorigenesis, and indeed early stage tumorigenesis, is not understood. Here we demonstrate that submucosal accumulation of mast cells in two

independent murine models of intestinal-type gastric cancer functionally contributes to tumorigenesis and the maintenance of established tumors and that this coincides with the presence of alternatively activated protumorigenic tumor-associated macrophages. We provide evidence that tumor cell-derived IL-33 stands on the apex of a cascade by which mast cells, through recruitment of tumor-associated macrophages and their support of a vascular network, ensures the growth and maintenance of gastric tumors. Indeed, ablating mast cells or macrophages in tumor-bearing mice either individually with cromolyn or clodrosomes, or together with PLX3397, were associated with a vascular collapse and tumor hypoxia. Importantly, we support the relevance of our functional findings in mice with a correlation in humans where a mast cell activation signature associates with poor patient outcome for intestinal-type gastric cancer.

Within the tumor microenvironment, myeloid cells are among the most important innate immune cells that promote tumor formation and restrict the effect of many forms of therapy, most notable immunotherapy[55]. In particular the plasticity of tumor-associated macrophages has attracted interest as therapeutic opportunities to limit the angiogenic and immune suppressive functions conferred by their alternatively activated endotype[3]. On the other hand, mast cells have been proposed to be an abundant source of VEGF, TGFβ, and other angiogenic factors, as well as matrix metalloproteinase 9 and related stromal remodeling enzymes[56]. Indeed, mast cells are one of the earliest cell types recruited into the tumor microenvironment[11,57], and progression of gastric cancer correlates with both the accumulation of chymase-positive mast cells and increased microvascular density[11,58]. Functionally, these observations are mirrored by an angiogenic switch proposed to be orchestrated by mast cells during early stages of pancreatic cancer in mice required for the extensive vascular elaboration to prevent hypoxia during expansion of tumors[16]. Surprisingly, the authors in this early study had argued against a role for macrophages in mast cell-dependent induction of early tumor angiogenesis, based on their observations that macrophage recruitment to pancreatic islets remained unaffected by mast cell deficiency. Indeed, our observations that IL-33 result in the induction of *Vegfa* expression in isolated mast cells supports this view and this is corroborated by reduced microvessel density in tumors of *gp130*$^{FF}$; *St2*$^{-/-}$ mice. However, our observations presented here following either genetic ablation of mast cells or macrophages, or their respective pharmacological inhibition by cromolyn and clodronate or PLX3397 strongly suggest that the endothelial cell support provided by mast cells also depends in part on the recruitment of macrophages.

Because we functionally link tumor growth to the presence of mast cells in two independent genetic models, it is highly unlikely that this arises from independent serendipitous effects of the two models other than mast cells. Thus, it is highly likely that the incompletely understood mechanisms by which cromolyn blocks mast cell degranulation mediated the anti-tumor effect observed here, rather than cromolyn also affecting monocytes, the production of the neutrophil product myeloperoxidase, and circulating growth[59]. Akin to tumor-associated macrophages being induced by tumor cells to adopt an alternatively activated endotype optimized for wound healing, neoplastic epithelia appears to also corrupt the wound healing function of mast cells to facilitate tumor growth[55]. Accordingly, tumor cell-specific overexpression of the master-regulator c-Myc not only results in epithelial hyperproliferation (when coinciding with increased expression of BH3 survival proteins) but also in expression of chemo-attracting CCL2, CCL3, and CCL5 and concomitant accumulation of mast cells[16]. Here we identify a second mechanism that depends on the

IL-1 cytokine family member IL-33, which acts as a chief regulator of innate immunity and inflammation, and maintains epithelial barrier functions of the intestine. Indeed, IL-33 serves *a bona fide* alarmin and is released by damaged epithelium as an endogenous danger signal to activate innate immune responses[28]. Accordingly, mast cells and ILC2s respond through the IL-33/ST2 axis to the damaged intestinal mucosa arising from acute injury or infection. Therefore mast cells are found at sites of epithelial recovery[60], and ILC2 cells release IL-13 to stimulate the intestinal stem cell compartment, respectively[61]. Intriguingly, our experimental lines of evidence from tumors of *gp130*$^{FF}$; *St2*$^{-/-}$ mice show that their reduced tumor burden correlates with decreased mast cell frequency rather than a change in ILC2 abundance. While not ruling out minor contributions by ILC2, regulatory T-cells or other IL-33 responsive cells, the increased tumor mass in *gp130*$^{FF}$; *St2*$^{-/-}$ mice following adoptive transfer of St2-proficient mast cells, provides definitive functional evidence that mast cells act as the major cell population through which IL-33 promotes tumor growth. Accordingly tumor growth of *gp130*$^{FF}$ mice was susceptible to the highly mast cell-selective cellular deficiency in *gp130*$^{FF}$; *Cpa3-Cre*; *Mcl1*$^{fl/fl}$ mice, or to cromolyn-dependent inhibition of mast cell degranulation. Meanwhile, and consistent with the capacity of IL-33 to promote tissue repair and wound healing, IL-33 can also mediate metaplasia of the gastric epithelium[62].

Although epithelial cells of barrier tissues exposed to the environment are major sources of IL-33, and IL-33 expression is further increased during inflammation and in the tumor epithelium[63], the exact nature of the factors that induce IL-33 expression and its release remain unclear. For instance TNF, IL-1β, TGFβ, prostaglandins, lipopolysaccharides, and other pathogen-associated molecular patterns have all been suggested as drivers for IL-33 expression during intestinal adenomatous polyposis[64–66]. Here, we provide evidence for IL-33 expression in gastric tumor epithelium to also be stimulated by an IL-11/ Stat3 signaling cascade, which we and others have identified as an absolute requirement for effective growth of gastrointestinal tumors. However, at this stage we can only speculate that subepithelial myofibroblasts and possibly tumor-associated endothelial cells, which have been proposed as sources of IL-11[67], may also contribute to epithelial expression of IL-33. Furthermore, because mast cells express gp130, we could not formally exclude the possibility that the presence of the hypermorphic *gp130*$^F$ allele in mast cells may promote their proliferation and survival[46], as this receptor allele augments Stat3 signaling in response to IL-6 family cytokines. However, we noted similar submucosal mast cell accumulation adjacent to tumors in *gp130*$^{FF}$ and in Tg(*Tff1-CreERT2*); *Pik3ca*$^{H1047R/+}$; *Pten*$^{fl/fl}$ mice, where the latter cohorts express WT gp130. Likewise, reconstitution of *gp130*$^{FF}$ mice with WT bone marrow did not affect mast cell numbers and tumor burden, as would be expected if the *gp130*$^F$ allele augmented mast cell activity.

Therapeutic inhibition of IL-33 signaling is gaining interest with anti-ST2 antibodies being trialed for asthma. Given the dichotomous activity of IL-33 in tumor biology, timing of anti-IL-33 signaling therapy will be crucial. In the context of gastric cancer, mast cells may not only promote metastasis but also modulate the immunosuppressive tumor environment through the release of IL-17[68]. Thus, IL-33 signaling provides a target to restrict the tumor promoting activities of the myeloid compartment and may ultimately enable rational combination therapies to alleviate the activity by which myeloid cells limit the antitumor immune response. The linear signaling cascade identified here comprising IL-11/IL-33/mast cells/macrophages/tumor cells should provide complementary molecular and cellular targets for the development of improved cancer therapies.

## Methods

**Study approval**. All animal studies were conducted in accordance with all relevant ethical regulations for animal testing and research including the Australian code for the care and use of animals for scientific purposes. All animal studies were approved by the Animal Ethics Committee of the Ludwig Institute for Cancer Research, the Walter and Eliza Hall Institute of Medical Research, or Austin Health.

We have complied with all relevant ethical regulations for work with human participants. Collection and usage of human gastric cancer tissues was approved by the Peter MacCullum Cancer Center Ethics Committee and informed consent was obtained from all subjects.

**Mice**. Knockin mice ($gp130^{Y757F/Y757F}$, alias $gp130^{FF}$ mice), the compound mutants strains $gp130^{FF}$; $Stat3^{+/-}$[39,40] and Tg($Tff1$-CreERT2); $Pik3ca^{H1047R/+}$; $Pten^{lox/+}$ as well as wild-type control mice were bred on a mixed C57B6 × 129/Sv background. $gp130^{FF}$; $c$-$Kit^{W-sh/W-sh}$[41], $gp130^{FF}$; $Csf1r^{-/-}$[69], and $gp130^{FF}$; $St2^{-/-}$ mice[70], knockout of the Il-33 receptor (encoding gene name: $Il1rl1$), $gp130^{FF}$; $Cpa3$-Cre; $Mcl1^{lox/lox}$[44] (mast cell-deficient $gp130^{FF}$ compound mutant mice) were maintained on a C57B6 background.

Tg($Tff1$-CreERT2); $Pik3ca^{H1047R/+}$; $Pten^{lox/+}$ compound mutant mice were established, where mutant allele induction is induced by the Tff1-CreERT2 transgene and is specific for the gastric epithelium of the stomach[71].

$gp130^{FF}$ mice used in the PLX-treatment experiment were propagated on a C57B6 background.

Cohoused, age- and gender-matched littermates were utilized for all experiments. All strains were housed under specific pathogen–free conditions.

**Human gastric cancer tissue**. Human gastric cancer tissue micro arrays (TMA) were used to identify toluidine blue-stained mast cells in gastric disease. TMA were established previously[72]. In short, formalin-fixed paraffin-embedded gastric tissue samples we selected, pathology was confirmed by independent pathology review and 0.6 mm punches were reembedded to generate the TMA. Usage of human gastric cancer tissues was approved by the Peter MacCullum Cancer Center Ethics Committee and informed consent was obtained from all subjects.

**Inhibitors and treatment regimes**. Cromolyn sodium salt (referred to as cromolyn; Sigma–Aldrich, Cat# C0399) is a clinically approved mast cell-degranulation inhibitor[73]. Cromolyn was dissolved in phosphate buffered saline (PBS) and administered at a concentration of 75 mg/kg body weight via the intraperitoneal route (100 µl) three times per week for 6 consecutive weeks. Control cohorts received PBS.

Clodronate, formulated as liposome-loaded clodrosomes (containing 5 mg clodronate per 1 ml of clodrosome suspension), was injected intraperitoneal as 50 µl clodrosomes suspension twice per week for 6 consecutive weeks. Liposomes containing suspension was injected as placebo in control mice.

The csf1r/c-kit/Flt3-specific inhibitor PLX3397 (Plexxicon), supplied at 800 mg PLX3397 per kg of chow, was given to mice for 4 consecutive weeks represents an estimated dose of 100 mg per kg body weight daily. Vehicle cohorts received unmanipulated chow. For the follow-up cohorts, mice were kept for 4 weeks on unmanipulated chow following their 4-week-treatment period.

**Tissue preparation and processing**. Tumors and adjacent antral tissues were resected and weighed and then snap-frozen for RNA or protein isolation. Entire stomachs were removed and fixed in 10% neutral buffer formalin for histological analysis[39].

**Histological and immunohistochemical analysis**. Hematoxylin-Eosin, toluidine blue, Alcian blue, Safranin O, and May-Grünwald-Giemsa staining of formalin-fixed paraffin-embedded stomach slides were performed according to theory and practice of histological technique from JD Bancroft[74].

For in vivo assessment of proliferation by anti-BrdU staining, tissues were collected 2 h after $i.p.$ injection of 50 mg/kg BrdU (Amersham Biosciences, GE Healthcare). Stainings for apoptosis (Cell Death Detection Kit, Roche) and tissue hypoxia (60 mg/kg HP1 $i.p.$ injection 30 min prior to tissue collection, detection with Hypoxyprobe-1 Kit, from Hypoxyprobe Inc., USA) were performed according to manufacturers' instructions.

All other immunostaining procedures were conducted as follows. For antigen-retrieval, paraffin-embedded sections were either heated in citrate buffer in a microwave pressure cooker (pH 6 for 15 min) or were incubated in 0.1% trypsin, 3% acetic acid solution at 37° for 10 min. Sections were then blocked in 10% (v/v) normal goat serum for 1 h at room temperature. Primary antibodies were diluted in 10% (v/v) normal goat serum and incubated overnight at 4 °C in a humidified chamber. Biotin-labeled secondary antibodies from the Avidin Biotin Complex ABC-kit (Vector Laboratories) were used according to the manufacturer's instructions. Visualization was achieved using 3,3-Diaminobenzine (DAB, DAKO). Images were generated with Aperio ImageScope v11.2.0.780 software. Primary antibodies used: rabbit anti-mouse CD31 (1:200 dilution, Abcam, Cat# 28364, RRID: AB_726362), rat anti-mouse F4/80 (1:200 dilution, Abcam, Cat# 6640, RRID: AB_1140040), rat anti-mouse B220 (1:200 dilution, BD Biosciences, Cat#

550286, RRID:AB_393581), rabbit anti-mouse CD3 (1:200 dilution, Abcam, Cat# ab5690, RRID:AB_305055), rat anti-mouse CD8 (1:150 dilution, eBioscience, Cat# 14–0808–82, RRID: AB_2572861), and rat anti-mouse Foxp3 (1:100 dilution, eBioscience, Cat# 12–5773–80, RRID: AB_465935).

**Immunofluorescence staining**. Paraffin-embedded stomach sections were dewaxed, rehydrated and antigen-retrieval was performed by heating in EDTA buffer (pH9) for 15 min in microwave pressure cooker. Section were incubated for 1 h at room temperature with goat anti-mouse IL-33 (1:150 dilution, R&D Systems, Cat# AF3626, RRID:AB_884269) and then for 1 h with AlexaFluor 568-conjugated donkey anti-goat secondary antibody (Molecular Probes Cat# A-11057, RRID: AB_142581). After counter staining with spectral DAPI (1:200 dilution, Perkin-Elmer, Cat# FP1490) for 5 min, sections were mounted with Vectashield mounting solution (VECTORlabs, Cat# H-1400). Imaging was preformed with a Vectra 3.0 Automated Quantitative Pathology Imaging System (PerkinElmer, Cat# CLS142338) and representative images were produced using Phenochart™ v1.0.4 software (PerkinElmer).

**Quantification**. For quantification of mast cell numbers at least three fields of view (10x ocular) of three toluidine blue section per mouse were counted, submucosal area was measured and data represented as mast cells number/area tissue. Similarly, for quantification of CD3, BrdU, and ApopTag staining, either entire tumors or three fields of view of three sections were counted and analyzed comparing to tissue area.

F4/80, CD31, B220, and Hypoxia immune-staining was quantified using Metamorph software (Molecular Devices) or Fiji scripts (ImageJ, https://Fiji.sc/) determining percentage positive area per area tissue.

**Isolation of gastric epithelial cells, immune cells, and flow cytometric analysis**. Glandular stomachs or antral tumors were dissected, cut into very small pieces and incubated at 37 °C for 30 min in $Ca^{2+}$- and $Mg^{2+}$-free HBSS medium plus 2.5% FCS and 1 mM EDTA with gentle shaking. Then samples were vortexed for 30 s and the supernatants containing intraepithelial lymphocytes were separated from the tissue fragments and kept on ice. The remaining tissue samples were further digested in Collagenase/Dispase (Roche) and DNase I (Roche) in $Ca^{2+}$- and $Mg^{2+}$-free HBSS medium plus 2% FCS for 45 min at 37 °C under continuous rotation. Samples were vortexed for 30 s once during incubation and once after the incubation. Afterwards, cell suspensions from both incubations were pooled, filtered and washed in PBS plus 5% FCS for analysis by flow cytometry.

Single cell suspensions were stained for cell surface markers (listed below) and cell viability was controlled with propidium iodide (ThermoFisher Scientific, Cat# P3566) or SYTOXBlue (ThermoFisher Scientific, Cat# S34857) staining. Cell sorting was performed with an Aria II cell sorter (BD Bioscience). All cell-type specific flow cytometric gating strategies are presented as a supplementary figure (Supplementary Fig. 8a–e). Mast cells sorting: from $CD45^+EpCam^-$ cell population the $CD11b^-$ cells were selected, then finally the $c$-$Kit^+FceRI^+$ population represents the mast cells. Macrophage sorting: from $CD45^+EpCam^-$ cells, the $F4/80^{High}CD11b^+$ population was selected and back-gated to confirm that the selected macrophage population was $Ly6C^-Ly6G^-$.

The following fluorochrome-conjugated antibodies were used for flow cytometric cell sorting and analysis: CD16/CD32 (1/100 dilution, clone 93, Cat# 14-0161-86), EpCAM-FITC (1/400 dilution, clone 9CA,Cat# 11-5791-82), Ly6c-eF450 (1/300 dilution, HK1.4, Cat# 48-5932-82), F4/80-PE-Cy7 (1/400 dilution, BM8, Cat# 25-4801-82), St2-PerCP-eFluor710 (1:200 dilution, clone RMST2-2, Cat# 46-9335-82), Foxp3-PE (1/200 dilution, clone FJK-165, Cat# 12-5773-82), Gata3-PE (1/200 dilution, clone TWAJ, Cat# 12-9966-42), FceR1-PE-Cy7 (1/300, clone 36951, Cat# 25-5898-82), and CD3-PE-Cy7 (1/1000, 145-2C11, Cat# 25-0031-82) all from Ebioscience; CD11b-PE (1/400 dilution, M1/70, Cat# 553311), Ly6g (1/300 dilution, 1A8, Cat# 560602) from BD Pharmingen and CD45.2-A700 (1/400 dilution, clone S450-15-2), CD4 (1/50 dilution, clone GK 1.5), CD11b-PB (1/400 dilution, M1/70), and c-Kit-APC (1/200, ACK-2) from WEHI monoclonal antibody facility.

**Isolation and stimulation of BMDM**. Bone marrow was collected from the femur and tibia of mice by flushing with sterile PBS. Cells were washed with PBS twice and were filtered through a 100 mm sieve. Resulting cell suspensions were cultured in DMEM supplemented with 10% (v/v) FCS and L929 conditioned medium for 7 days with fresh media changes every second day until fully differentiated into BMDM. After reseeding of cells, alternative macrophage polarization was conducted by stimulation with IL-4 (20 ng/ml, Preprotech) and IL-13 (20 ng/ml, Preprotech) for one day.

**Cytokine bead array**. A total of 50,000 isolated gastric $gp130^{FF}$ mast cells were cultured in 96-well plates for 3 h in 30 ng/mL IL-33. Supernatants were collected, diluted 1:2 in assay buffer and assayed for cytokine concentration using the Bio-Rad Bioplex cytokine bead assay (Bio-Rad Mouse 23-Plex Panel M60009RDPD) strictly according to manufacturer's instructions.

**BMMC adaptive transfer assay**. Mast cells were generated from bone marrow from either *gp130FF; ST2+/+* or *gp130FF; ST2−/−* mice. Bone marrow was extracted and cells were maintained in IL-3-containing culture medium for several weeks until >95% pure BMMC cultures were established. Purity of BMMCs (FceR1+, c-Kit+) was determined via flow cytometry. For the BMMC transplantation assay, syngeneic BMMC from the same colony as the host mice were used. Either *gp130FF; ST2+/+* or *gp130FF; ST2−/−* BMMCs were injected into the tail vein of *gp130FF; ST2−/−* mice. Each mouse received $10^7$ BMMC at 5, 8, and 11 weeks of age and final tumor analysis was performed at 14 weeks of age.

**Gastric organoid assay**. Antral tumors from *gp130FF* mice were used to establish tumor-epithelial organoids. Organoids were established and maintained in IntestiCult™ Organoid Growth Medium (StemCell Technologies) according to manufacturer's protocols. Established organoids were stimulated with 100 ng/ml IL-11 or PBS control for 4 h and then processed for gene expression analysis via qPCR. For assessment of organoid growth, PBS or IL-11 stimulated organoids were monitored over 4 days.

**RNA isolation and quantitative RT-PCR**. Total RNA was extracted from frozen tissue samples using Trizol®Reagent (life technologies, Cat# 15596026) and cDNA was prepared from 2 μg RNA using the High capacity cDNA Reverse Transcription kit (Applied Biosystems, Cat# 4368813) according to the manufacturer's protocol. From isolated gastric mast cells and from FACS sorted macrophages, RNA was extracted using the RNeasy Plus Micro Kit (QIAGEN, Cat# 74034) and cDNA synthesis was performed with the ThermoScript™ RT-PCR System (Invitrogen, Cat# 11146-024) according to the manufacturer's instructions.

Quantitative RT-PCR analyses were performed in technical triplicates with SensiMix SYBR kit (Bioline, Cat# QT605–20) using the *ViiA™ 7 Real Time PCR* System (life technologies). Further details and the sequences of the used oligonucleotides are described in the supplementary methods and Supplementary Table S1.

**Kaplan–Meier survival analysis**. All Kaplan–Meier survival analysis was performed with KMplot (KMplot.com)[53]. Data from 179 intestinal-type gastric cancer patient were analyzed and following settings were applied: probes = all probe sets, Laurens classification = intestinal-type, Data sets included: GSE14210, GSE15459, GSE22377, GSE29272, and GSE51105, but excluding GSE62254 (as recommended; due to markedly different survival and shifted expression compared to the other data sets). All other settings were kept as default.

**Statistics**. Unless otherwise stated, data are presented as mean values ± standard error of the mean (M ± SEM). Exact *n* values are depicted for every data set in the figure legends and they are always true biological replicates, not technical replicates. All data sets were tested for normality with Shapiro–Wilk normality test. Comparisons between two mean values were performed by two-tailed unpaired Student's *t*-test using Prism6 software (GraphPad Software, California, USA). When data sets had significantly different standard deviations according to Prism's *F*-test of variances, Welch correction was applied. *T*-test *t* values and corresponding degree of freedom (df) are given in figure legends. For comparisons of more than two groups one-way ANOVA test was performed with multiple comparison correction (either Tukey or Bonferroni). ANOVA *F* values are depicted in each figure legend as F (DFn, Dfd), where DFn is the df nominator and Dfd the df denominator. *P* values of less than 0.05 were considered statistically significant.

**Sample size determination**. G*power3.1 software was used to estimate the minimal required sample sizes. Based on previous mean values and SD from previous studies using the *gp130FF* mouse model, we calculated a minimal sample size for eight mice per group, when the mean difference is 40% and the SD 20% (unpaired two-tailed *t*-test; α = 0.05, Power = 0.95). Due to difference in availability and frequency of target genotypes born, the number of mice analyzed between cohorts differed. Sample size for subsequent analysis of tissue section staining was limited to number available, as tissues were required for several downstream processes including FACS analysis, RNA and protein expression in addition to histological and immunohistochemical analysis.

**Randomization**. No formal randomization procedure was performed. However, for genetic mouse experiments, littermates were used to compare different target genotype mice. For in vivo treatment experiments, different treatment groups were equally distributed between mouse litters and mouse cages as well as female and male ratio were kept at 1:1.

**Data exclusion**. No data was excluded after commencement of quantification. In a few cases, tissue slides were excluded from downstream analysis due to strong background staining or absence of tumor tissue. These slides were excluded from analysis before commencement of quantification and without knowledge of genotypes/treatment details (aka blinded).

**Blinding**. Assessors were blinded for determining tumor masses and numbers and for quantification of histo- and immunohistochemical staining. Blinding was achieved through labeling minimized to mouse IDs. Genotypes and treatment were only linked back to the mouse ID after completion of quantification.

**Reporting summary**. Further information on research design is available in the Nature Research Reporting Summary linked to this article.

## Data availability

The Stat3 chromatin immunoprecipitation-sequencing (ChIP-Seq) data have been deposited in the NCBI gene expression omnibus (GEO) under the accession code GSE48285. The source data underlying Figs. 1b, d, 2b–h, 3b–e, 4b–e, 5a–c, 6a, c, d, f, 7a–g, and 8a and Supplementary Figs 1b, 2a, b, g, i, 3d, 5a, b, g, h, and 6e, f are provided as a source data file. All the other data supporting the findings of this study are available within the article and its supplementary information files and from the corresponding author upon reasonable request. A reporting summary for this article is available as a Supplementary Information file.

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

## Acknowledgements

We thank the animal facility staff of La Trobe University and the Ludwig Institute for Cancer Research as well as the Austin and WEHI histology department for technical assistance. We also acknowledge the excellent support for cell sorting by David Baloyan (ONJCRI) and image generation and analysis by Cameron Nowell (Monash University) and Lachlan Whitehead (WEHI). This work was supported through the Victorian State Government Operational Infrastructure Support, the Nation Health and Medical Research Council (NHMRC) of Australia grants 1092788, 1067244, 1069024 and Cancer Council Victoria's Grant-in-Aid 1160708. M.E. is a NHMRC Principal Research Fellow and also received funding from Ludwig Cancer Research.

## Author contributions

Experimental work: M.F.E, A.J., M.B., R.J.J.O'D., T.P., C.D., J.B., S.T., E.T., N.D.H., N.E., A.R.P., and F.M. Data interpretation: M.F.E, A.J., M.B., R.J.J.O'D., A.B., M.L.H., M.A.G., F.M., and M.E. Writing and editing of manuscript: M.F.E., M.L.H., F.M., and M.E. Study concept: M.F.E., A.J., F.M., and M.E. Funding: M.E.

## Additional information

**Competing interests:** The authors declare no competing interests.

