## [Peer Review File · Nature Communications]

Reviewers' comments:

Reviewer #1: Expert in Cancer immunology (Mast cells)
(Remarks to the Author):

This is an interesting manuscript which contains a substantial amount of data concerning the roles of mast cells and IL-33 in models of gastric cancer. The use of a number of complementary tumour models is a strength and the focus on the interaction between IL-33 and mast cells in regulating macrophage populations is of interest.

1. Unfortunately assessments of mast cell dependency were limited to the Wsh mice which have a number of additional defects. An additional non-ckit dependent model would be an asset since several other IL-33 responsive cell populations are also modified in number in this mast cell deficient strain.
2. Macrophage depletions and CCR2 deficiency is well known to reduce tumour growth. The authors should reference some of these earlier studies and also consider the role of monocytic MDSC in more detail. The classification of macrophage subsets is also too simplistic given the recognized complexities of the tissue environment.
3. It would be more convincing for their proposed model of mast cell dependency if the authors addressed the impact of a defect in ST2 on mast cell exclusively (e.g. through Bone marrow mast cell reconstitution of mast cell deficient mice)
4. The impact of mast cell and IL-33 on tumor angiogenesis should be considered in more detail, both experimentally and in the discussion.
5. The role of IL-33 in modulating the activities of ILC2 cells and Treg has been largely ignored in the current analysis and proposed model
6. The human "MC signature" proposed should include classical mast cell mediators/ indicators of mast cell activation and transcripts such as the FcER1 beta chain. Many cell types could be responsible for the mediators being examined currently
7. CSF2 and IL-6 are not chemokines- as indicated in the abstract
8. Some of the studies appear to have been completed in very few mice (e.g. n=3) with no indication that these or other studies have been repeated.
9. The justification for using t-tests for some of the data sets is unclear. Have all of these data passed a normality test?

Reviewer #2: Expert in Cancer immunology (IL-33)
(Remarks to the Author):

Eissmann et al's paper entitled "IL-33-mediated mast cell activation promotes gastric cancer through macrophage mobilization" reveals that IL-33, mast cells and TAM promote tumorigenesis in the gp130^{FF} model of GC. They also show that IL-33 can activate mast cells to produce cytokines and chemokines that are potentially involved in TAM differentiation and function. They additionally demonstrate that ST2 deficiency results in decreases in mast cells in tumor submucosa and mast cell deficiency leads to a reduction in the number of TAM. They conclude that IL-33-mediated mast cell activation promotes gastric cancer through macrophage mobilization. Although their data are largely consistent with the conclusion, they are mostly correlative. The finding is not entirely novel because it has been shown that the enhancing effect IL-33 on intestinal polyposis in the Apc(Min/+) model is associated with increased numbers of mast cells (PMID: 25918379) and mast cells are crucial for polyps formation in this model (PMID: 18077429). In the same vein, the IL-33/M2 axis in tumorigenesis has already been well established. The mechanistic studies are superficial. How mast cells recruit M2 is not convincingly established. Nonetheless, the paper does provide new insights into the pathogenesis and potential new therapeutic approach for gastric

cancer.

The following are specific critiques:

In the abstract, the statement "release of the macrophage attracting chemokines Csf2, Ccl3, and Il6" is incorrect. IL6 and CSF2 are not chemokines and CCL3 is not exclusively a chemokine for macrophages.

In the introduction part, the authors stated "Interleukin (IL)33 is a danger associated signal that can serve as a molecular "alarmin" when released upon necroptotic and necrotic cell death including that of cancer cells.". This statement is not supported by actual research papers. The author should reference the research papers (not review papers) actually supporting this statement.

Figure 1. needs further clarification. Do tumors arise only in the gastric antrum in this model since mast cell accumulation seems to occur only in this area? Why did mast cells only accumulate this area? What are the target cells of the gp130 signaling? If target cells of gp130 signaling are not mast cells, why mast cells accumulated in this area?

Figure 2. Stating "mast cells indeed assumed a "driver function" is no appropriate because mast cell accumulation is secondary to the driver gp130 mutation (page 6 last paragraph).

Figure 3. The characterization of T cells is cursory. Alternatively, the cause for the reduced tumor burdens in ckit^{wsh} mice could be attributed to increased antitumor cell-mediated immune responses. Since CD8 and Treg are also the target cells of IL-33, inclusion of IHC assays for CD8 and foxp3 can help clarifying this issue.

Figure 4. It is not clear whether these treatments or genetic deficiency are specific for M2 but not for other myeloid cells.

Figure 5. when conclude, authors state that "tumor derived IL-11 induces Il33 gene expression in gastric tumor cells". However, there is no data showing that IL-11 is tumor derived.

Figure 7. For Kaplan-Meyer survival analysis, it is not clear why the GSE62254 dataset was excluded. Also, is the expression level of IL-33 associated with OS? Is expression level of IL-33 positively correlated with signatures of mast cells or M2?

Reviewer #3: Expert in Gastric cancer
(Remarks to the Author):

The authors present a tremendous amount of data regarding the role of mast cells in gastric neoplasia.

Data are here summarized with some questions marked... The authors use a Gp130^{ff} mouse model with deficient shp2 binding to STAT3 leading to increased IL6/IL11 dependent STAT3 activity and gastritis. In this model, they see increased mast cells at regions of tumors more than normal tissue. The use of a STAT3 +/- normalized Mast cell numbers. They also used a model which uses a TFF1 CreER in combination with both PIK3CA inducible H1047R and floxed PTEN (an unusual gene combination). Tumors in these mice also yield mast cells. In all cases mast cells are submucosal rather than intratumor. In human tumors they see submucosal mast cells in intestinal tumors but not in diffuse or in gastritis/IM. For experiments they used a variant of mice with an altered c-kit promoter leading to reduced expression of stem cell factor and reduced mast cells. These mice had smaller tumors with reduced endothelial cells. Figure d also shows a large increase

in hypoxia with the altered Kit allele.

Question 1-- Can the authors comment on this finding and clearly tie it to the CD31 findings?

Treatment with cromolyn to reduce mast cell degranulation reduce proliferation, macrophages and angiogenesis and tumor mass. They did bone marrow rescue with GP130FF with BM of normal mice so that the mast cells would have WT GP130 and got similar results (except for higher mast cells) indicating that the effects are not due to the GP130 manipulation in the mast cell/hematologic population. This is a nice experiment.

Further experiments showed a reduction in macrophages but not CD3 or B220, used as a marker of B-cells, leading the authors to hypothesize that mast cells contribute to macrophages.

Question 2: Could the reduction of macrophages merely be due to less tumor formation leading to less tissue damage? This is for comment, not for experiment.

Additional studies show that with both the GP130 and Pik3ca/pten model that macrophages are seen. The findings with macrophages in the mucosal tissue in sup 3d are clearly suggestive of some difference.

Question/comment 3: The statement on the bottom of page 8 that data suggest a contribution of macrophages to tumor formation are overstated. These data just show their presence.

The authors then use CSF1r null mice. They report a few remaining mice but that these have no tumors.

Question 4: Can the authors give precise data and report how long the CSF1r null mice lived? Is the tumor assessment at similar ages?

The authors also showed reduction in tumors with clodronate to target macrophages. They claim the clodronate does not impact mast cells but the data show some reduction.

Question 5: Could additional slides/samples be quantified for mast cells to determine to what extent this occurs? Also, this begs the question if there was any change in mast cell density with the CSF1R KO mice with the GP130ff?

Question 6: Can the authors comment on the interpretation of the finding of activated arg1/fizz1/mrc1 but not Nos2?

Comment/question 7: The text on the bottom of page 10 that this proves that the alternatively activated endotype is the rate limiting step for establishing tumor microvasculature and tumor growth. For example, the authors have not done experiments to compare macrophages with/without this endotype. I am not asking the authors to do this experiment but just to temper conclusions.

Question 8: is there any relationship to the IL-33 signal staining as shown by 5b to location of mast cells?

Question 9: Can the authors quantify the lack of growth difference that is shown in Figure 5e?

The authors then make IL-33 signaling deficient mice with an st2 KO.

Question 10: Can the authors comment on effects of ST2 ko other than effects upon IL-33?

Further, these mice show reduced macrophage mediators. They authors assess other cell

populations impacted by IL-33, finding comparable numbers of ILc2 and TREGs.

In figure 7 the authors see survival effects with the mast cell signature in intestinal cancers.

Question 11: Can the authors be sure this is not confounded by other clear variables. For example, if the mast cell signature is less in MSI tumors and MSI tumors have a better prognosis, the signature would show worse survival.

12. Comment for discussion: Their data were primarily on tumor formation. Does that necessarily mean that targeting mast cells will be a target for established cancers?

Detailed response to issue raised by **Reviewer 1** (paraphrased in blue)

1. Unfortunately assessments of mast cell dependency were limited to the *Wsh* mice which have a number of additional defects. An additional non-*ckit* dependent model would be an asset since several other IL-33 responsive cell populations are also modified in number in this mast cell deficient strain.

We have obtained the *Cpa3-Cre;Mcl1^{f/f}* mice as the generally accepted “gold standard” for a mast cell deficient mouse model. We now show in our revised **Fig. 2e** that the total tumor burden (i.e. mass and number) is significantly reduced in mast cell-deficient *gp130^{FF};Cpa3-Cre;Mcl1^{f/f}* mice compared to their mast-proficient littermate controls. We also now present new data (**Fig. 6g**) from the converse experiment where *gp130^{FF};St2^{-/-}* mice reconstitute with in vitro amplified *gp130^{FF};St2^{+/+}* mast cells show increased tumor burden compared to these hosts reconstitute with congenic *gp130^{FF};St2^{-/-}* mast cells. Collectively, these additional observations provide compelling strong functional evidence that the tumor promoting activity conferred through IL-33/St2 signaling depends on mast cells.

2. Macrophage depletions and *CCR2* deficiency is well known to reduce tumour growth. The authors should reference some of these earlier studies and also consider the role of monocytic MDSC in more detail. The classification of macrophage subsets is also too simplistic given the recognized complexities of the tissue environment.

On page 9, macrophages have been acknowledged as drivers of tumor growth and as therapy targets and two references have been added. While we agree with this reviewer that the classification of macrophages is more complex, we believe that the simplified classification used here is sufficient to highlight the role of M2-like tumor-associated macrophages to constitute part of the St2/mast cell/macrophage cascade by which IL-33 promotes gastric tumorigenesis.

Regarding a potential role of monocytic MDSCs in our models, we have assessed the abundance of monocytic and granulocytic CD45+ MDSCs in tumors of *gp130^{FF}* mice either proficient or deficient for St2 expression (see below, **Reviewer Fig 1**). Curiously, we find an increase of either MDSC type in the smaller tumor we observe in *gp130^{FF};St2^{-/-}*. Because of the acknowledged challenges to manipulate either of these MDSC population independently *in vivo*, we have elected not to pursue the functional consequence for our finding further and therefore present the data only for the benefit of this reviewer below.

3. It would be more convincing for their proposed model of mast cell dependency if the authors addressed the impact of a defect in *ST2* on mast cell exclusively (e.g. through Bone marrow mast cell reconstitution of mast cell deficient mice)

We agree with the reviewer and now present data that reconstitution of *gp130^{FF};St2^{-/-}* mice with *in vitro* expanded mast cells from *gp130^{FF};St2^{+/+}* mice increases tumor burden when compared to hosts of the same genotype reconstituted with *gp130^{FF};St2^{-/-}* mast cells (new **Fig. 6e**). We also show that in

the $gp130^{FF};St2^{+/+}$ reconstituted hosts approx. 8% of submucosal toluidine-blue mast cells stain positive for St2 expression (**Fig. S6f**), consistent with the slow turnover of resident ($St2^{-/-}$) mast cells in these hosts and the fact that the transferred $gp130^{FF};St2^{+/+}$ mast cells are competing with the resident $St2^{-/-}$ mast cells, which already occupied compartments in the $gp130^{FF};St2^{-/-}$ hosts at time of the cell transfer.

4. The impact of mast cell and IL-33 on tumor angiogenesis should be considered in more detail, both experimentally and in the discussion.

We now provide data to show a direct connection between IL-33, mast cells and angiogenesis. Specifically, we have added a qPCR graph to Fig.5d showing that *Vegfa* expression is elevated in isolated mast cells upon IL33-stimulation. Also, updated Fig. 6c now shows submucosal CD31 quantification in addition to the tumor analysis. Reduced submucosal microvessel density in ST2-deficient tissue could be either a direct effect of smaller number and activity of mast cell in the tumor-adjacent submucosa or indirectly through the reduced number of F4/80 macrophages. We discuss this now in greater details on page 18 in the Discussion section.

5. The role of IL-33 in modulating the activities of ILC2 cells and Treg has been largely ignored in the current analysis and proposed model

It is important to acknowledge ILC2 and Treg as targets for IL-33 signaling and hence to consider contribution of these cells to the anti-tumor effect we observed in $gp130^{FF};St2^{-/-}$ mice. Given our focus on mast cells in this manuscript, we elected to conduct the adoptive mast cell experiment to unambiguously show that a majority of the anti-tumor effects observed in the absence of St2 are mediated by mast cells. However, these observations do not preclude that other IL-33 responsive (immune) cells also contribute to the reduced tumor burden in $gp130^{FF};St2^{-/-}$ mice. This is now acknowledged and discussed in more details in the Discussion section on p19 of the revised manuscript.

6. The human “MC signature” proposed should include classical mast cell mediators/ indicators of mast cell activation and transcripts such as the *FcER1* beta chain. Many cell types could be responsible for the mediators being examined currently

The conjecture of the reviewer’s comment is that inclusion of a classical mast cell marker in our “MC signature” would add to the specificity argument of our proposed signature. We have constructed separate survival curves for a combination of the 3 mast cells genes *KIT*, *FCER1G* and *HDC* (encoding Histidine Decarboxylase) (new **Supplementary Fig. 7a**) and these 3 markers combined with our MC-signature (see below, **Reviewer Fig 2**). While all of these survival plots suggest worse survival outcomes with elevated expression, only our MC-signature reaches statistical significance. Given

that these 3 genes are also expressed in cells other than mast cells (i.e. *KIT* in melanoblasts and various hematopoietic stem cells; *FCER1G* in platelets, monocytes, dendritic cells and neutrophils; *HDC* in basophils), we believe that our proposed MC-signature correlates survival with the effect of genes expressed primarily in mast cells.

7. CSF2 and IL-6 are not chemokines- as indicated in the abstract

The abstract has been corrected as suggested.

8. Some of the studies appear to have been completed in very few mice (e.g. n=3) with no indication that these or other studies have been repeated.

Although all results depicted in our original submission have been reproduced in at least one independent repeat, we have now clarified this information for each figure independently. Moreover, we have increased the number of independent observations/samples included for our revised analysis of **Figures 1a, 2d, 4e, 5a, 5c, 6c, 6d**, and **Supplementary Figure 2g**. Finally, the legend to **Supplementary Figure 2g** was corrected as the number of independent samples (n) was erroneously given as 3 rather than the correct range 4-7.

9. The justification for using t-tests for some of the data sets is unclear. Have all of these data passed a normality test?

All data sets have been tested and passed the Shapiro-Wilk test for normal distribution as provided in the Prism 7 software. The only exception being the tumor data set obtained from our newly included adoptive mast cells experiments (**Fig. 6g**), where a Mann-Whitney U test was performed. In addition, we also show in **Supplementary Fig. 6f** this data after performing ROUT outlier analysis. Further information regarding statistics has been added in the Methods section and in the individual figure legends.

Detailed response to issue raised by **Reviewer 2** (paraphrased in blue)

The finding is not entirely novel because it has been shown that the enhancing effect IL-33 on intestinal polyposis in the Apc(Min/+) model is associated with increased numbers of mast cells (PMID: 25918379) and mast cells are crucial for polyps formation in this model (PMID: 18077429).

Our survey of the literature on the proposed roles of IL-33 in solid malignancies indicates highly conflicting and controversial findings, with limited evidence to predict outcomes in one solid cancer based on findings from another one. Furthermore, there are also outcomes contradicting the findings referred to in PMID: 25918379 within intestinal tumors, where we and others have shown increased tumor burden in St2-deficient back grounds (Malik et al., JCI 126:4469; Eissmann et al., Cancer Immunol Res 9:404). Thus, we argue strongly that findings presented here are novel.

In the same vein, the IL-33/M2 axis in tumorigenesis has already been well established. The mechanistic studies are superficial.

While the M2 macrophage involvement in gastric tumor growth is indeed well established, our manuscript identifies an entire, novel signaling cascade emanating from IL-11 induced expression of IL-33 in cancer cells, to IL-33-mediated activation of tumor-associated mast cells, to macrophages recruitment, leading to increased angiogenesis and tumor growth. We strongly disagree with this reviewer's comment about the superficial nature of our study. For instance, we use Chip-Seq analysis to present strong mechanistic evidence for IL-11-dependent transcription of IL-33 (**Supplementary Fig. 5i,j**) and IL-11-dependent stimulation of gastric tumor organoids (**Fig. 5e,f**). Likewise, we use chemokine gene and protein expression analysis (**Fig. 5c,d**) to demonstrate IL-33 mediated mast cells activation, and complement this analysis by showing that these effects are dependent on St2 expression on mast cells (**Fig. 6d**). Finally, our mast cell adoptive transfer experiment provides definitive evidence that the St2-mediated signaling within mast cells confers pro-tumoral activity (**Fig. 6g**).

How mast cells recruit M2 is not convincingly established. Nonetheless, the paper does provide new insights into the pathogenesis and potential new therapeutic approach for gastric cancer.

We would like to clarify that we do not propose that mast cells (or their soluble mediators) affect the polarization of tumor-infiltrating macrophages towards an M2-endotype. Indeed, there is plenty of evidence in the literature that it is the tumor cells that "corrupt" tumor associated macrophages to adopt an M2 endotype through immune cell-, and tumor cell-derived soluble factors. We therefore designed our experiments to functionally correlate mast cell abundance with the abundance of macrophages positive for the marker F4/80, which shows comparable expression on M1 and M2 polarized macrophages. Fig. 4h shows that the macrophages present within the *gp130^{ff}* gastric tumors are skewed towards a M2 phenotype, but we do not claim that mast cells drive this M2 polarization. However, we have now included new data from macrophage migrations assays (**Supplementary Fig. 5g**), which demonstrates the chemotactic effect of the mast cell-secreted factors on macrophages *in vitro*.

In the abstract, the statement "release of the macrophage attracting chemokines Csf2, Ccl3, and Il6" is incorrect. IL6 and CSF2 are not chemokines and CCL3 is not exclusively a chemokine for macrophages.

This has been corrected.

In the introduction part, the authors stated "Interleukin (IL)33 is a danger associated signal that can serve as a molecular "alarmin" when released upon necroptotic and necrotic cell death including that of cancer cells.". This statement is not supported by actual research papers. The author should reference the research papers (not review papers) actually supporting this statement.

We have included two additional primary research papers in support of the necrosis-related IL-33 release. Additionally, it has been noted that cell-death independent IL-33 release can occur, which we also refer to with two references.

Figure 1. needs further clarification. Do tumors arise only in the gastric antrum in this model since mast cell accumulation seems to occur only in this area? Why did mast cells only accumulate this area? What are the target cells of the gp130 signaling? If target cells of gp130 signaling are not mast cells, why mast cells accumulated in this area?

In the $gp130^{FF}$ model the first lesions develop in the antrum and later on we also observe smaller lesions in the cardia of the stomach. Accordingly, we have focused our analysis of mast cell density on the more prominent antral tumors. However, we also observed increased density of mast cells in the cardia of $gp130^{FF}$ mice prior to the development of tumors, which later on is further increased in the submucosa adjacent to cardiac tumors (data not shown).

Figure 2. Stating "mast cells indeed assumed a "driver function" is no appropriate because mast cell accumulation is secondary to the driver gp130 mutation (page 6 last paragraph).

As per the reviewer's suggestion, this statement has now been modified to: "...mast cells indeed promote gastric tumor growth, .."

Figure 3. The characterization of T cells is cursory. Alternatively, the cause for the reduced tumor burdens in ckitwsh mice could be attributed to increased antitumor cell-mediated immune responses. Since CD8 and Treg are also the target cells of IL-33, inclusion of IHC assays for CD8 and foxp3 can help clarifying this issue.

As per the query, we have now performed immunohistochemical stainings for CD8 and Foxp3 and show in the revised version of our manuscript representative images (**Fig. 3a**) and corresponding quantifications (**Fig. 3b**). We find no significant difference in density of CD8⁺ or Foxp3⁺ cells between tumors of mast cell proficient $gp130^{FF};c-Kit^{+/+}$ versus mast-deficient $gp130^{FF};c-Kit^{W-sh/W-sh}$ mice.

Figure 4. It is not clear whether these treatments or genetic deficiency are specific for M2 but not for other myeloid cells.

We (Poh et al., Cancer Cell 31:563) and many others have shown that the Csf-1R deficiency restricts the development of macrophages prior to their polarization towards M1 and M2 endotypes. Others have shown that clodronate liposome treatment reduces the abundance of the macrophage marker F4/80, which is expressed in M1 and M2 polarized cells and that this is associated with reduced tumour burden in many other models (Wu et al., J Invest Dermatol 134:2814).

Figure 5. when conclude, authors state that "tumor derived IL-11 induces Il33 gene expression in gastric tumor cells". However, there is no data showing that IL-11 is tumor derived.

We have previously characterized in great details the origin of IL-11 in the $gp130^{FF}$ gastric cancer model (Putoczki et al., Cancer Cell 24:257). This analysis revealed that approx. 50% of IL-11 mRNA associated with gastric tumors is contributed by EpCAM⁺ epithelial cells.

Figure 7. For Kaplan-Meyer survival analysis, it is not clear why the GSE62254 dataset was excluded. Also, is the expression level of IL-33 associated with OS? Is expression level of IL-33 positively correlated with signatures of mast cells or M2?

Data set GSE62254 has been excluded from the Kaplan-Meyer survival analysis as recommended by the KMplot analysis tool. This is due to the markedly different patient survival within this data set compared to all other gastric cancer data sets included in the analysis. We have performed the Kaplan-Meyer survival analysis for *IL33*, its receptor and the combinations with our IL33/mast cell activation signature. Surprisingly, high *IL33* expression is associated with better survival in the analyzed data sets (HR=0.63, logrank P=0.022) (see below **Reviewer Fig 3**). However, the IL33 receptor gene *IL1RL1* is associated with worse survival (HR= 1.57, logrank P=0.049) (new Supplementary Fig. 7b). Neither the combination of *IL33* + IL33/mast cell activation signature (HR=1.3, logrank P=0.22) nor the combination of *IL33* + alternative activation macrophage signature

(HR=0.83, logrank P=0.48) show significant correlation with survival (see below **Reviewer Figure 3**). Therefore, *IL33* expression is a less well-suited indicator for clinical outcome. Our study suggests, that it is important to identify receptor expression, the cell-types responding to IL-33 and the factors produced by the IL-33 activated cells.

Detailed response to issue raised by **Reviewer 3** (paraphrased in blue)

Question 1-- *Can the authors comment on this finding and clearly tie it to the CD31 findings?*

Our previous analysis of the $gp130^{FF}$ gastric cancer model (Putoczki et al., Cancer Cell 24:257) suggests a close correlation between IL-11 expression (which drives the growth of tumors in this model), phospho-Stat3 (which provides intracellular signaling in IL-11 responsive cells) and CD31 expression in these tumors. We now show that genetic or pharmacological interference with mast cells correlates with decreased CD31 expression and increased hypoxia. Collectively, these correlative observations argue very strongly for an involvement of CD31+ endothelial cells.

Question 2: *Could the reduction of macrophages merely be due to less tumor formation leading to less tissue damage? This is for comment, not for experiment.*

We take it as a conjecture from the reviewer's proposed mechanism that less tissue damage would reduce the level of IL33 produced/released. Such a mechanism could indeed be contributing as a compounding negative regulatory "feed-back loop", however, such a scenario still would require an initial genetic event that reduces the abundance of infiltrating macrophages.

Question/comment 3: *The statement on the bottom of page 8 that data suggest a contribution of macrophages to tumor formation are overstated. These data just show their presence.*

The reviewer is correct, that the data presented on p8 is of a correlative nature only; for this reason we carried out the subsequent experiments on mice where we genetically or pharmacologically ablate macrophages, using $Csf-1R^{KO}$ mutant and clodronate-treatment respectively.

Question 4: *Can the authors give precise data and report how long the CSF1r null mice lived? Is the tumor assessment at similar ages?*

In our hands the average life span of $gp130^{FF};Csf1r^{-/-}$ mice is approximately 80 days. The littermate controls used in this experiments were of similar age when analyzed (i.e. 88.2d for $gp130^{FF};Csf1r^{+/+}$ mice and 90d for $gp130^{FF};Csf1r^{-/-}$ mice, respectively). We have added this information in the legend for **Fig. 4b**.

Question 5: *Could additional slides/samples be quantified for mast cells to determine to what extent this occurs? Also, this begs the question if there was any change in mast cell density with the CSF1R KO mice with the GP130ff?*

As requested, we have analyzed additional histology slides from stomach of PBS-control and clodronate-treated mice to produce additional data points that we have included in the revised **Fig. 4e** in order to have similar numbers of observations for both cohorts. Statistical analysis of this new data suggests that there is no significant difference. In addition, we have quantified mast cell-density in stomachs of $gp130^{FF};Csf1r^{-/-}$ mice and compared it to that of $gp130^{FF};Csf1r^{+/+}$ littermates; again this analysis reveals no significant difference between the cohorts. This data has been added in new panel **Fig. 4c**.

Question 6: *Can the authors comment on the interpretation of the finding of activated arg1/fizz1/mrc1 but not Nos2?*

In mouse macrophages, expression of *Arg1*, *Fizz1* and *Mrc1* are validated and widely used markers to assess M2 polarization, while *Nos2* is considered a marker for M1 macrophages. The observation presented in **Fig. 4g** showing increased expression of *Arg1*, *Fizz1* and *Mrc1* and very low *Nos2* expression is therefore suggesting preferential macrophage polarization towards an M2 endotype in gastric tumor tissues.

Comment/question 7: *The text on the bottom of page 10 that this proves that the alternatively activated endotype is the rate limiting step for establishing tumor microvasculature and tumor*

growth. For example, the authors have not done experiments to compare macrophages with/without this endotype. I am not asking the authors to do this experiment but just to temper conclusions.

As requested by the reviewer, we have now “tempered” our conclusions by stating: “Given the alternative activated endotype of tumor-associated macrophages, our data collectively suggests that their abundance is likely to be a rate-limiting factor for the establishment of tumor microvasculature and the growth of gastric tumors.”

Question 8: *is there any relationship to the IL-33 signal staining as shown by 5b to location of mast cells?*

Our analysis depicted in **Fig. 5b** localizes IL-33 producing cells to the epithelial cell compartment of the tumors as well to adjacent cells in the submucosa. We would not expect for strong IL-33 staining to co-localize with mast cells, because we only expect limited physical association of IL-33 with mast cells through the few hundred St2 receptor molecules expressed on each of these cells. These may not only be below the detection limit of our immunohistochemical method, but detection may also be further hampered by the overlapping epitopes in IL-33 for binding to St2 and the anti-IL-33 detection antibody.

Question 9: *Can the authors quantify the lack of growth difference that is shown in Figure 5e?*

As requested by the reviewer, we have now quantified the effect of IL11 on organoid growth depicted in the micrographs of **Fig. 5e** and present the data in new **Supplementary Fig. 5h**.

Question 10: *Can the authors comment on effects of ST2 ko other than effects upon IL-33?*

It is important to clarify whether St2 receptor deficiency impacts on cytokines other than IL-33, given the shared use of one of the corresponding receptor subunits. However, St2 provides the IL-33 *ligand specific subunit*, while the second subunit (IL-1RAcP) is shared among several IL-1 family members (i.e. IL-1 α , IL-1 β , IL-33 and IL-36; Guenther et al., *Immunity* 47:510; Jo et al., *Cytokine* 83:33). Therefore, analysis of St2 knock-out mice, or cells derived thereof, specifically assess the lack of signaling in response to IL-33.

Question 11: *Can the authors be sure this is not confounded by other clear variables. For example, if the mast cell signature is less in MSI tumors and MSI tumors have a better prognosis, the signature would show worse survival.*

The reviewer identifies a very relevant issue not only for our analysis, but for all published Kaplan-Meier analyses. However, addressing this issue in the context of constructing KM plot based on gene expression pattern is difficult for at least two reasons: Biologically, gene signatures indicative of MSI (or indeed any of the other four major gastric cancer subtypes) are not unambiguous. Technically, the online Kaplan-Meier software analysis tool does not allow filtering for specific gene sets while testing simultaneously for the proposed mast cell signature. In order to perform the suggested analysis, a relative complex bioinformatic study would be required, which is beyond the scope of our study here.

Question 12: *Comment for discussion: Their data were primarily on tumor formation. Does that necessarily mean that targeting mast cells will be a target for established cancers?*

The reviewer rightfully points out that our data obtained from our models of genetic deficiency for mast cells (*c-Kit^{W-sh}* and *Cpa3^{Cre};Mcl1^{fl}*), macrophages (*Csfr^{KO}*) and St2 (*St2^{KO}*) does not allow us to address whether established tumors continue to depend on these functions and/or cell types. For this reason, we have complemented our study with a series of therapeutic experiments on tumor-bearing mice, incl. treatment with cromolyn (to inhibit mast cell degranulation), clodronate (to deplete macrophages), or with PLX3997 (to inhibit c-kit and c-fms to simultaneously reduce mast cells and macrophages). Consistent with published data on the ongoing requirement for mast cell-dependent angiogenesis for established pancreatic tumors (Soucek et al., *Nature Med* 13:1211), all

our treatment experiments revealed a growth inhibiting effect on established tumors. Conversely, the adaptive transfer of St2 positive mast cells into tumor-bearing *gp130^{F/F};St2^{-/-}* hosts further increased their tumor burden (**new Fig. 6g**). Collectively, this data argues strongly that the IL-33/St2/mast cell/macrophage signaling axis remains required for established tumors and provides a therapeutic susceptibility for these established cancers.

Reviewers' comments:

Reviewer #1 (Remarks to the Author):

The authors have addressed this reviewers concerns very effectively. While there are still some small details upon which I may disagree the authors have done their best to clarify and provide the necessary additional data to overcome any major scientific issues with their work

Reviewer #2 (Remarks to the Author):

The authors have addressed most of my concerns. There are still some issues that need to be clarified.

1. What are the target cells of the gp130 signaling in their model of gastric cancer?
2. The authors seemed to arbitrarily draw the conclusion that IL33 expression is a less well-suited indicator for clinical outcome despite their own analysis shows that high IL33 expression is associated with better survival. Similar results have also been reported in the literature by other groups in multiple human cancers. The authors need to thoroughly review papers showing this kind of results and provide thoughtful discussion about the underlying basis of protumor and antitumor activities of IL-33.

Reviewer #3 (Remarks to the Author):

I have no further comments and am satisfied with the revision.

Detailed response to issues raised by **Reviewer 2** (paraphrased in blue)

The authors have addressed most of my concerns. There are still some issues that need to be clarified.

1. What are the target cells of the gp130 signaling in their model of gastric cancer?

In the *gp130^{FF}* mutant mice all cells carry the mutation and therefore all cells expressing the gp130 signaling machinery will be hyper responsive to ligand mediated activation of the pathway. The revised manuscript text on page 8 has been modified to make this point clearer. The modified sentence is now highlighted red.

2. The authors seemed to arbitrarily draw the conclusion that IL33 expression is a less well-suited indicator for clinical outcome despite their own analysis shows that high IL33 expression is associated with better survival. Similar results have also been reported in the literature by other groups in multiple human cancers. The authors need to thoroughly review papers showing this kind of results and provide thoughtful discussion about the underlying basis of pro-tumor and antitumor activities of IL-33.

This reviewer is commenting on our previous response to his/her earlier question. We agree with this reviewer that our observation that IL-33 expression is associated with better survival (which was a reviewer only Figure in revision 1) warrants discussion.

Indeed, akin to our data provided in the response to this reviewer on IL-33 expression and gastric cancer survival, there are other studies in various solid tumors showing that IL-33 expression correlates with improved survival (Yang M *et al.*, PloS One 13:e0193428; Rössle M *et al.*, Int J Surg Pathol 24:394; Koster R *et al.*, Int J Cancer 142:1594). However, the opposite correlation has been described as well (Zhang *et al.*, Int J Neurosci 127:210; Fang M *et al.*, Cancer Res 77:2735, Wang Z *et al.*, Tumour Biol 37: 11127; Tong X *et al.*, Mol Oncol 10:113; Chen SF *et al.*, J Pathol 231:180. Meanwhile, others did not find significant association between survival and IL-33 expression (Cui G *et al.*, Cancer Immunol Immunother 64:181; Hu W *et al.*, Pathol Oncol Res 23:615). These differences might be due to multiple factors. Several studies used immunohistochemistry to detect protein levels and others analyzed RNA expression, while some investigated IL-33 cytokine abundance in the serum (Hu LA *et al.*, Asian Pac J Cancer Prev 14:2563; Hu W *et al.*, Oncotarget 23:35123). In addition to these technical differences, there are biological differences between tumor types, and there could also be variances within histological tumor sub-types, tumor stage and other patient factors affecting the functional consequences of IL-33 expression. Furthermore, simply assessing IL-33 expression in the tumor might not always directly reflect receptor-mediated signaling, because of the detection of intracellular IL-33 not yet released into the microenvironment. In light of these caveats, and consistent with our mouse tumor experiments demonstrating a ST2-mediated pro-tumoral effects, we suggest that assessing the expression levels of ST2 and the associated IL-33/mast cell activation signature may prove to be a more accurate predictor than IL-33 expression. However, this requires further validation.

REVIEWERS' COMMENTS:

Reviewer #2 (Remarks to the Author):

I am ok with the answers.